

# The major stratospheric final warming in 2016: Dispersal of vortex air and termination of Arctic chemical ozone loss

Gloria L. Manney[1,2] and Zachary D. Lawrence[2]

[1]NorthWest Research Associates, Socorro, New Mexico, USA.
[2]Department of Physics, New Mexico Institute of Mining and Technology, Socorro, New Mexico, USA.

*Correspondence to:* Gloria L Manney (manney@nwra.com)

**Abstract.** The 2015/16 Northern Hemisphere winter stratosphere appeared to have the greatest potential yet seen for record Arctic ozone loss. Temperatures in the Arctic lower stratosphere were at record lows from December 2015 through early February 2016, with an unprecedented period of temperatures below ice polar stratospheric cloud thresholds. Trace gas measurements from the Aura Microwave Limb Sounder (MLS) show that exceptional denitrification and dehydration, as well as
5   extensive chlorine activation, occurred throughout the polar vortex. Ozone decreases in 2015/16 began earlier and proceeded more rapidly than those in 2010/11, a winter that saw unprecedented Arctic ozone loss. However, on 5–6 March 2016 a major final sudden stratospheric warming ("major final warming", MFW) began. By mid-March, the mid-stratospheric vortex split after being displaced far off the pole. The resulting offspring vortices decayed rapidly preceding the full breakdown of the vortex by early April. In the lower stratosphere, the period of temperatures low enough for chlorine activation ended nearly a
10  month earlier than that in 2011 because of the MFW. Ozone loss rates were thus kept in check because there was less sunlight during the cold period. And although the winter mean volume of air in which chemical ozone loss could occur was as large as that in 2010/11, net chemical ozone loss was considerably less.

We use MLS trace gas measurements, as well as mixing and polar vortex diagnostics based on meteorological fields, to show how the timing and intensity of the MFW and its impact on transport and mixing halted chemical ozone loss. Our
15  detailed characterization of the polar vortex breakdown includes investigations of individual offspring vortices and the origins and fate of air within them. Comparisons of mixing diagnostics with lower stratospheric $N_2O$ and middle stratospheric CO from MLS (long-lived tracers) show rapid vortex erosion and extensive mixing during and immediately after the split in mid-March; however, air in the resulting offspring vortices remained isolated until they disappeared. Although the offspring vortices in the lower stratosphere survived longer than those in the middle stratosphere, the rapid temperature increase and
20  dispersal of chemically-processed air caused active chlorine to quickly disappear. Furthermore, ozone-depleted air from the lower stratospheric vortex core was rapidly mixed with ozone rich air from the vortex edge and midlatitudes during the split. The impact of the 2016 MFW on polar processing was the latest in a series of unexpected events that highlight the diversity of potential consequences of sudden warming events for Arctic ozone loss.



# 1   Introduction

Sudden stratospheric warmings (SSWs), which are characterized by abrupt warming and weakening or reversal of the polar wintertime westerly circulation (e.g., Andrews et al., 1987, and references therein), lead to extreme variability in Northern Hemisphere (NH) polar temperatures, as well as in the structure, evolution, and intensity of the Arctic stratospheric polar vortex. SSWs are in part responsible for the smaller potential for ozone loss in NH than in Southern Hemisphere (SH) spring (e.g., Andrews, 1989; WMO, 2014). SSWs are relatively common in the NH, occurring at a rate of $\sim$0.6 events per year by many common definitions (see, e.g., Butler et al., 2015, and references therein). However, recent studies have shown that SSWs affect Arctic lower stratospheric chemical ozone loss in ways much more complex than a simple association of low (high) temperatures with more (less) ozone loss (Manney et al., 2015a, b, and references therein). Thus, understanding the complex relationships between SSW dynamics, stratospheric vortex evolution, and chemical composition and processing, is critical to diagnosing and predicting ozone loss and recovery in the Arctic and its climate consequences.

Recent Arctic winters with SSWs have led to different extremes in polar processing and ozone loss: The 2012/13 NH winter was exceptionally cold in December, but a major vortex-split SSW in January gave rise to two unusually strong offspring vortices that moved far into sunlight (Manney et al., 2015a). The combination of extensive polar processing activity prior to the SSW and ample sunlight exposure following the SSW led to the earliest onset of rapid Arctic ozone loss in the 12-year record from the Aura Microwave Limb Sounder (MLS); that loss continued through the end of January (when the polar vortex dissipated completely). In contrast, polar processing was effectively halted in the NH winter of 2014/15 by a brief minor SSW in early January (Manney et al., 2015b). Although the minor SSW had similar signatures to a major SSW in the middle and upper stratosphere, it left the lower stratospheric vortex virtually unscathed except for causing temperatures to rise above chlorine activation thresholds for a couple of weeks. This resulted in anomalously little chlorine activation, and the highest wintertime ozone seen in the twelve-year MLS record.

Interannual variability in NH winters is also reflected in the timing of the springtime stratospheric final warming. These events mark the transition of the stratospheric winter circulation from westerly to easterly, where it remains until the following autumn. Numerous studies suggest that the timing of final warmings is related to SSWs earlier in winter: Labitzke (1982) showed that final warmings following major SSWs in January or February in the 1950s through 1970s were usually delayed due to late winter cooling after the SSW; recently, Hu et al. (2014) showed a statistically significant relationship between midwinter (December through March) major SSWs and late ($\sim$April and May) final warmings. The converse is also true in that many early final warmings tend to occur in winters without a prior strong SSW (Labitzke, 1982; Hu et al., 2014, and references therein). The end of any potential for polar processing and chemical ozone loss typically closely follows the final warming, as temperatures rise above activation thresholds and the breakdown of the polar vortex rapidly disperses chemically processed air, both of which hasten chlorine deactivation (e.g., Prather and Jaffe, 1990; Tan et al., 1998; Santee et al., 2008, and references therein). Because of this interplay of chemical/microphysical and dynamical processes, the abruptness and timing of the final warming plays a substantial role in polar processing, and there is large interannual variability in the evolution of final warmings (e.g., Black and McDaniel, 2007). Labitzke (1982) first noted that SSWs in late February or March often turn



directly into final warmings. Such an early and abrupt final warming, initiated by a major SSW early enough in the season that recovery is possible, but after which recovery does not occur, is referred to as a major final warming (MFW) (see, e.g. Hoffmann et al., 2002; Labitzke, 2002; Naujokat et al., 2002; Manney et al., 2006a, b; Blume et al., 2012). MFWs can result in more rapid mixing of air from the polar vortex than later, more gradual final warmings (e.g., Waugh and Rong, 2002; Akiyoshi

and Zhou, 2007), and rapid cessation of ozone loss (e.g., Konopka et al., 2003; Marchand et al., 2004; Manney et al., 2006b).

As we will show below, the 2015/16 Arctic winter was the coldest on record (since at least 1979) in the lower stratosphere through January. Minimum temperatures in the lower stratosphere were far below those in the 2010/11 winter/spring when extensive chemical loss led to record low values of Arctic ozone in April 2011 (Manney et al., 2011; WMO, 2014, and references therein). There was thus the potential for extreme chemical ozone loss that might have exceeded that in 2011.

However, an MFW beginning in early March 2016 resulted in the breakup of and dispersal of chemically processed air from the vortex, which halted chemical loss much earlier than in 2011. We show that the critical factor resulting in less ozone loss than in 2011 was the early final warming in 2016, presenting another instance when the occurrence of a major SSW (in this case an MFW) played a key role in determining the amount of ozone loss in an Arctic winter, in a way differing from the diverse scenarios we have already observed in recent years.

In this paper, we analyze meteorological data from the MERRA-2 (Modern Era Retrospective-analysis for Research and Applications) reanalysis and trace gas data from the Aura MLS instrument to give an overview of dynamical conditions and chemical composition in the polar vortex during the 2015/16 winter, and detail the effects of the MFW that shattered the vortex in early March 2016, which curtailed polar processing and limited chemical ozone loss. We focus on transport and mixing during the vortex breakup and its effects on the composition of air that was dispersed from the vortex. A comprehensive picture

of the vortex evolution and breakup is obtained using a newly developed package for characterizing multiple vortices. We describe the evolution of the vortex and trace gases through the MFW and associated vortex splitting, focusing on mixing and dispersal of chemically processed air from the vortex.

After describing the datasets and methods used (Section 2), Section 3 provides an overview of the dynamical conditions and chemical composition of the vortex throughout the 2015/16 winter. Section 4.1 then provides an overview of the evolution of

25 trace gases and relationships to bulk diagnostics of mixing and transport barriers. In Section 4.2, the synoptic/regional processes leading to these relationships are diagnosed. Section 5 gives a summary and conclusions.

## 2 Data and Methods

### 2.1 MERRA-2 Reanalysis

The National Aeronautics and Space Administration's (NASA) Global Modeling and Assimilation Office (GMAO) MERRA-2

dataset (Bosilovich et al., 2015) is a global reanalysis that covers the satellite era from 1980 to the present. It uses the Goddard Earth Observing System version 5.12.4 assimilation system with a cubed-sphere model to perform its analyses. As in its predecessor, MERRRA, an incremental analysis update (IAU) (Bloom et al., 1996) applies the analysis tendency gradually over the 6-hour analysis window. MERRA-2 contains substantial upgrades over MERRA, including new input data, model



constraints, and parameterizations (Molod et al., 2015; Takacs et al., 2016). Assimilated fields are provided on a $0.625° \times 0.5°$ longitude/latitude grid with 72 hybrid $\sigma$-pressure levels. Here we primarily use the wind, temperature, and potential vorticity fields provided in the "M2I3NVASM" file collection (Global Modeling and Assimilation Office (GMAO), 2015), the set of dynamically consistent fields obtained after the IAU step; these fields are provided at the full model resolution at 3-hour intervals (8 times per day).

## 2.2 MLS Data

The Earth Observing System (EOS) Aura satellite was launched in July 2004, in a 98° inclination orbit that provide data coverage from 82° S to 82° N latitude on every orbit. Aura MLS measures millimeter- and submillimeter-wavelength thermal emission from the limb of Earth's atmosphere. Detailed information on the measurement technique and the Aura MLS instrument is given by Waters et al. (2006). Vertical profiles are measured every 165 km along the suborbital track and have a horizontal resolution of ∼200–500 km along-track and a footprint of ∼3–9 km across-track. In this study we use version 4 (v4) Aura MLS $N_2O$, $HNO_3$, $H_2O$, $HCl$, $ClO$, and $O_3$ measurements from Arctic winters spanning 2004/05 through 2015/16. The quality of these data is described by Livesey et al. (2015a). Vertical resolution is about 2.5–3 km for $O_3$, 3 km for $H_2O$, HCl, and ClO, 3–5 km for $HNO_3$, and 5–6 km for $N_2O$ in the lower to middle stratosphere, and about 5 km for CO in the middle stratosphere. Single-profile precisions are approximately 0.03–0.1 ppmv, 0.2–0.3 ppbv, 0.1 ppbv, 0.6 ppbv, 13–20 ppbv, and 16 ppbv for $O_3$, HCl, ClO, $HNO_3$, $N_2O$, and CO, respectively, and 5–15% for $H_2O$. The v4 MLS data are quality-screened as recommended by Livesey et al. (2015a). For daily maps, MLS data are gridded at 2°latitude by 5°longitude using a weighted average around each gridpoint of 24 hours of data centered at 12:00 UT.

Equivalent latitude (EqL, the latitude that encloses the same area between it and the pole as the corresponding potential vorticity, PV, contour, Butchart and Remsberg, 1986) and scaled PV (sPV, scaled as in Dunkerton and Delisi, 1986; Manney et al., 1994) are used in the analysis described below. These quantities, as well as temperatures from MERRA-2, are obtained at MLS locations from an updated version of the MLS derived meteorological products (DMPs) described by Manney et al. (2007). MLS data are interpolated to isentropic surfaces using temperatures from MERRA-2.

## 2.3 Vortex and temperature diagnostics

To investigate the potential for polar chemical processing and ozone loss during the 2015/16 winter, we use a standard set of polar processing diagnostics calculated from MERRA-2 data. We primarily make use of diagnostics described by Lawrence et al. (2015), including minimum temperatures ($T_{min}$), the volume of air with temperatures below polar stratospheric cloud (PSC) existence thresholds as a fraction of vortex volume ($V_{PSC}/V_{Vort}$), maximum PV gradients, and the area of the polar vortex in sunlight (or sunlit vortex area). All of these diagnostics are calculated from the 12:00 UT temperature and potential vorticity fields provided by MERRA-2 interpolated to isentropic surfaces between 390 and 580 K (approximately 120 to 30 hPa, or 14 to 24 km). Further discussion of these diagnostics and their significance to polar chemical processing can be found in Manney et al. (2011) and Lawrence et al. (2015).



Our analysis makes use of a detailed characterization of the 2015/16 stratospheric polar vortex, particularly during the period of time when the vortex split into multiple offspring. We use the CAVE-ART (Characterization and Analysis of Vortex Evolution using Algorithms for Region Tracking) analysis package, which was developed to comprehensively describe the state of the polar vortex throughout the winter season. A paper describing the full details and implementation of CAVE-ART is

5 in preparation (Lawrence and Manney, 2016); in short, CAVE-ART uses image processing and region tracking algorithms to objectively identify any number of vortex regions and track their positions through time. CAVE-ART identifies vortex regions based on altitude-dependent contours of sPV that we specify as being representative of the vortex edge. These sPV values are selected using climatological profiles of sPV spanning 25 isentropic levels between 390 and 1800 K to identify the sPV value at each level that coincides best with maximum sPV gradients from the MERRA reanalysis. Once CAVE-ART identifies

individual vortex regions, it filters out those below a specified area threshold; except where otherwise noted, we use herein an equivalent latitude threshold of 84°, which is an area of roughly 0.5% of a hemisphere. All remaining regions are then tracked through time using the full time resolution of the meteorological data (eight times per day for MERRA-2) until the regions fall below the area threshold, or in some cases merge with another region. CAVE-ART also calculates and saves diagnostics at every timestep that describe the position, size, and strength of each region. These diagnostics include 2-D moment diagnostics

such as aspect ratios and centroids (e.g., Matthewman et al., 2009; Mitchell et al., 2011), vortex areas, average altitudes, and vortex-edge windspeeds.

Such detailed characterizations are particularly useful during vortex-split SSW events wherein the resulting offspring vortices can vary in size and strength in ways that ultimately influence polar processing. For example, a preliminary version of CAVE-ART was used by Manney et al. (2015a), to show how the early January 2013 vortex-split SSW was responsible for

accelerated ozone loss in January 2013. For the current paper, the CAVE-ART characterization is particularly important because, as will be shown, during the 2016 MFW, the vortex rapidly weakened and briefly split into three offspring vortices at some levels. The capability to track more than two offspring vortices is, to our knowledge, currently unique to CAVE-ART, as other methods in the literature rely on moment diagnostics that can, at best, delineate between only two regions (Mitchell et al., 2011). We have included a supplementary animation that shows the evolution of the polar vortex during the March 2016

MFW, which illustrates the CAVE-ART characterization of the vortex split.

### 2.4 Transport and Mixing Diagnostics

EqL time series of MLS data are produced using a weighted average of MLS data in EqL and time, with data additionally weighted by measurement precision (e.g., Manney et al., 1999, 2007). All vortex averages of MLS data shown use the altitude-dependent sPV values derived for CAVE-ART to identify the edges of the vortex or vortices (Section 2.3). For averages in

multiple vortices, the sPV from the MLS DMPs is first used to determine whether the MLS measurement location is within any vortex. Those points that are within a vortex are then marked with the labels for individual regions provided by CAVE-ART to identify which of multiple vortices they are inside. Vortex averages are shown here for "bulk" (all MLS measurements with sPV greater than the altitude-dependent threshold), "sum" (the sum of all the regions with area greater than the 84°EqL cutoff used in the CAVE-ART runs), and for individual vortices identified by CAVE-ART. Averaging improves MLS precisions to



values smaller by a factor of about 10 to more than 100 over the single-profile precisions listed in Section 2.2 for EqL gridded
and vortex averaged fields..

sPV gradients and effective diffusivity ($K_{eff}$) as a function of EqL calculated from MERRA-2 PV are shown as "global" (that
is, characterizing amounts averaged around EqL contours) diagnostics of mixing. Gradients of sPV as a function of EqL provide
a measure of the strength of the vortex edge as a transport barrier averaged over each day and all vortices (Manney et al., 2011,
2015b, and references therein). $K_{eff}$ is expressed as log-normalized equivalent length, i.e., the length of a tracer contour with
respect to the contour of minimum length that would enclose the same area; high (low) values thus reflect complex (simple)
structure in tracer (here PV) contours and indicate strong (weak) mixing (e.g., Nakamura, 1996; Haynes and Shuckburgh,
2000; Allen and Nakamura, 2001). The magnitudes of $K_{eff}$ values depend strongly on the resolution of the PV fields used
in the calculations, but $K_{eff}$ distributions from MERRA-2 agree morphologically with those calculated from other analyses
and reanalyses. Similarly to sPV gradients, the gradients of long-lived trace gases on isentropic surfaces as a function of EqL
indicate the strength of the vortex edge transport barrier. We use the EqL/time gridded MLS fields to calculate these gradients.

The diagnostics of mixing and transport barriers described above represent averages around EqL contours, and thus give
information on bulk mixing properties; for example, the strength of the transport barrier at the EqL of the vortex edge is
an estimate of that barrier averaged over the the entire length of the edges of all vortices present at that time. To examine
regional mixing (e.g., variations along the edge of a vortex, or differences between individual vortices), we use the function
M (hereinafter referred to as $M$) to give a synoptic picture of the strength of the vortex transport barrier prior to, during,
and after the 2016 MFW. $M$ is a Lagrangian diagnostic calculated from parcel trajectories that has been used to highlight
processes related to transport and mixing in geophysical fluid flows (Mendoza and Mancho, 2010; de la Cámara et al., 2012;
de la Cámara et al., 2013; Smith and McDonald, 2014). The formal definition of the function $M$ is as follows: consider a point
in an $n$-dimensional space defined at an initial time $t_0$ by the general coordinates $(x_{1,0}, ..., x_{n,0})$. If a parcel is initialized at
this point and advected by the background velocity field ($\frac{dx_i}{dt}$), then the function $M$ at this point is then defined by the integral
equation $M(x_{1,0}, ..., x_{n,0}, t_0) = \int_{t_0-\tau}^{t_0+\tau} \left[ \sum_{i=1}^{n} \left( \frac{dx_i}{dt} \right)^2 \right]^{1/2} dt$. This is the Euclidean arc length of the trajectory traced out by
the parcel in the time interval $[t_0 - \tau, t_0 + \tau]$. If a grid of such points and parcels is constructed, then a field of $M$ can be
defined by calculating the above integral for each point within the grid. For our application, we instead calculate a field of $M$
by using zonal and meridional winds in a trajectory code to advect parcels initialized on a regular longitude/latitude grid. We
then calculate $M$ by summing up the distances between parcel locations at successive times assuming that these locations are
connected by great circle arcs.

We use the core of the Lagrangian trajectory diagnostic code described by Livesey et al. (2015b) to calculate parcel advection
using a fourth-order Runge Kutta scheme. The trajectories we use here are calculated via integrations with a fifteen minute
timestep from MERRA-2 winds. We initialize parcels on a 1.25 x 1.00 degree longitude/latitude grid (a grid spacing twice the
size of the MERRA-2 native grid) poleward of 20°N. The calculations of $M$ are based on isentropic trajectories that are carried
out for 15 days forward and backward (i.e., $\tau = 15$ days, for 30 days total) from 00:00 UT on the initialization date.

$M$ has been used before to study transport in Earth's polar vortices. de la Cámara et al. (2012) used $M$ to define hyperbolic
trajectories and invariant manifolds in the lower stratospheric flow of the southern hemisphere during the 2005 Southern





spring, which helped to explain transport across the vortex edge. In a later study, de la Cámara et al. (2013) used $M$ and reverse domain filling calculations of potential vorticity to diagnose signatures of Rossby wave breaking in the Antarctic polar vortex and explain the trajectories of isopycnic balloons launched as part of the Vorcore and Concordiasi field campaigns. Smith and McDonald (2014) used average values of $M$ and the area of large $M$ values to describe polar vortex strength and the permeability of its edge. Although there has been some recent debate on the usefulness of $M$ for describing flow manifolds and hyperbolic trajectories (see, e.g., Ruiz-Herrera, 2015; Balibrea-Iniesta et al., 2016), here we use $M$ in an arguably simpler manner similar to that of Smith and McDonald (2014); namely, large values of $M$ indicate parcels that were effectively trapped in a transport barrier for most of the trajectory timeline, whereas small values of $M$ indicate the opposite with parcels that were more prone to mixing. We also use an area diagnostic similar to that in Smith and McDonald (2014), obtained by calculating the area enclosed by contours of $M$ for the entire grid, and expressing these as an equivalent latitude, which we denote by "$M$-EqL". Although EqL is most commonly used to describe the area within PV or tracer contours (e.g., Butchart and Remsberg, 1986; Allen and Nakamura, 2003) (as PV-based EqL is used herein), we have found that examining $M$ and $M$-EqL together facilitates understanding of how the size, strength, and sharpness of the vortex edge transport barrier evolve with time.

We also use the trajectories described above to explore the origins and fate of air from the polar vortex during its breakup. We use the CAVE-ART identification of vortex regions to "tag" parcels that were initialized inside each valid and distinct vortex region. This allows us to examine the full history of parcels with respect to their original confinement within materially separated vortex regions. Similar trajectories were calculated using the full 3D trajectory code with diabatic motions, and only extending backwards from 12:00 UT each day, for use in reverse domain filling (RDF) calculations initialized with MLS data; the MLS fields used for these initializations are the gridded map fields described above.

## 3 Overview of 2015/16 Polar Vortex Evolution and Composition

Figure 1 gives an overview of dynamical conditions in the Arctic lower stratosphere during the 1979/1980 through 2015/16 winters using the polar processing diagnostics calculated from MERRA-2 as described above (Section 2.3). Figure 2 shows vortex-averaged (summed over all vortices identified by CAVE-ART) trace gases in the lower stratosphere from Aura MLS data for 2004/05 through 2015/16. The period of the Aura mission has included numerous winters with conditions at the extremes of Arctic variability, in both meteorology and chemical composition: The 2010/11 winter was not overall characterized by record low temperatures, but rather by an exceptionally prolonged period, extending into early April, of temperatures low enough to activate chlorine (Figure 1a,b, Figure 2d,e) and an unusually strong and persistent lower stratospheric vortex (Figure 1c) (Manney et al., 2011; WMO, 2014, and references therein). These conditions enabled unprecedented ozone loss (Figure 2f); in fact, the vortex remained strong and relatively cold after the period shown here, and ozone continued to drop, reaching a minimum of ∼1.5 ppmv in late April (e.g. Manney et al., 2011; WMO, 2014). While the vortex was exceptionally strong and vortex-averaged ozone loss unprecedented in the 2010/11 winter/spring, the size of the vortex during much of the winter (through late February), and the portion of it exposed to sunlight, were both less than average (Figure 1d).



In early January 2013, temperatures abruptly rose far above those at which chlorine can be activated (and hence chemical ozone loss occur) during one of the strongest "vortex-split" SSWs on record (Manney et al., 2015a); however, the exceptional cold prior to that event (Figure 1a,b), and exceptional exposure of the vortex (offspring vortices) to sunlight in December (January) (Figure 1d) led to denitrification comparable to that in 2011 (Figure 2b) and the largest early winter chlorine activation

and ozone loss on record (Figure 2d,e,f) (Manney et al., 2015a).

2014/15 provided a counterexample, where a very brief minor SSW, after which temperatures quickly dropped below the chlorine activation threshold again, led to rapid chlorine deactivation and, combined with exceptionally strong descent (as seen in the record $N_2O$/$H_2O$ decrease/increase, Figure 2a,c) (Manney et al., 2015b), the highest January/February ozone values in the MLS record.

In comparison to these previous recent years with exceptional combinations of dynamical conditions leading to unanticipated extremes in Arctic polar processing, the 2015/16 winter stands out as yet another unexpected extreme in variability of the Arctic winter stratosphere. Minimum temperatures (Figure 1a) were well below average from late November through mid-March, and near or at record lows from late December though January. The period of over a month, from late December through early February, with temperatures below the ice PSC threshold was unprecedented in the Arctic, where the previous record

rarely shows more than a few contiguous days below this threshold and never more than about three weeks (Figure 1a and Figure 3c,d). The long period of temperatures below the ice PSC threshold led to a much greater degree of dehydration than has been seen before in the Arctic: Compare the evolution of $H_2O$ in 2016 in Figure 2c with the small decrease in $H_2O$ seen in 2011 when there were separated periods in late January and February of about one and three weeks' duration, respectively, with temperatures below the ice PSC threshold. The presence of large ice PSCs also can lead to greater denitrification than

the presence of (typically smaller) NAT or liquid PSC particles alone (e.g., Wofsy et al., 1990; Hintsa et al., 1998; Santee et al., 1998; Lowe and MacKenzie, 2008; Dörnbrack et al., 2012; Wohltmann et al., 2013), and is consistent with the extreme denitrification evident in Figure 2b. Figure 1d shows that the 2015/16 vortex was not only larger than usual, but also had a larger area than usual receiving sunlight during January through mid-March.

The exceptionally cold conditions resulted in extensive early winter chlorine activation, with low HCl values in late Decem-

25 ber/early January matched only by those in 2012/13 (Figure 2d). Greater than usual sunlight exposure also resulted in high ClO (Figure 2e), with values in January through mid-February higher than those in 2011. Because of this extensive chlorine activation, chemical ozone loss began early, with a downward trend in vortex-averaged ozone (Figure 2f) seen beginning at the end of December 2015. Only in 2012/13, with exceptional cold and sunlight exposure in December (Figure 1a,d; Manney et al., 2015a), did ozone begin decreasing earlier. The onset of an observed ozone decrease means that chemical loss had become

large enough to dominate over replenishment by descent of air within the vortex; Figure 2a shows that early winter descent was similar in 2015/16, 2012/13, and 2010/11: thus the relative timing of the start of the ozone decrease reflects that of the onset of chemical loss.

Ozone continued to decrease in the vortex at a rate slightly faster than that in 2011 until the beginning of March 2016. If uninterrupted, ozone values would have been expected to drop lower than those in 2011 by mid-March. Instead, before mid-

35 March, a brief increase of about 0.5 ppmv was followed by about a week of decreasing values and then slightly increasing



values for the rest of the winter (Figure 2f). (At lower altitudes, near 430 K, not shown, vortex averaged ozone in 2016 was up to ∼0.2 ppmv lower than that in 2011 from late January until early March when the MFW began, but then rapidly rose above 2011 values.) Figures 1a and b show that temperatures rose above the activation threshold by mid-March, and Figures 1c and d show a sudden decrease in vortex strength and area just before mid-March, with the vortex being nearly gone by mid-

April. Furthermore, $N_2O$ values (Figure 2a), which had been decreasing steadily via descent in the vortex, rose suddenly by about 20 ppbv around the time of the vortex split (second red vertical line), subsequently dropped again by over 20 ppbv, then remained highly variable with little trend for the rest of the winter. The return of ClO values to near-zero was concurrent with the beginning of the final upturn in ozone in the vortex (Figures 2e,f). Temperatures began rising and ClO decreasing in the last week of February, with minimum temperatures exceeding the activation threshold just before the vortex split (Figure 1a).

The steep decrease in ClO began nearly a month earlier than that in 2011.

$V_{NAT}/V_{Vort}$ ($V_{PSC}/V_{Vort}$ calculated for nitric acid trihydrate PSCs and integrated over a winter) is a diagnostic commonly used to indicate the overall potential for polar processing and ozone loss (e.g., Rex et al., 2004; Tilmes et al., 2006). The polar processing potential in 2015/16 estimated using this diagnostic (Figure 3a) was nearly identical to that in 2010/11. Another measure, the number of days integrated over the lower stratosphere on which activation can occur (Figure 3b; e.g., Manney

et al., 2011, Supplementary Information) shows values similar to the largest previously observed excepting 2011. Similar diagnostics for the volume and duration of air below the ice PSC threshold (Figure 3c,d) indicate much greater potential for dehydration (and denitrification) than in any previously observed Arctic winter. It may be argued that more extensive denitrification in 2015/16 enhanced the ozone loss potential because of its effect of slowing chlorine deactivation (e.g., Douglass et al., 1995; Santee et al., 1996, 2008; Waibel et al., 1999; Davies et al., 2002). Thus, the critical factor resulting in less ozone loss

than in 2011 was the much earlier increase in temperatures and vortex breakup in 2016.

Figure 4 shows the vertical extent of polar processing and progression of the vortex breakup in 2015/16. The downward tilt of $N_2O$ contours in Figure 4 until early March indicates very regular descent within the vortex, as does the downward progression of $H_2O$ and $O_3$ contours above about 600 K where those species are not affected by lower stratospheric chemical processing. Increases in $N_2O$ throughout the domain, and in $O_3$ above ∼600 K, after the MFW began suggest increased mixing

into the vortex. The disappearance of significant vortex regions is marked by blank regions, and occurs in late March at and above about 850 K (in the middle stratosphere), early April down to about 500 K, and after mid-April at levels below that.

In contrast to 2011 and 2013, during which evidence of renitrification was seen above 400 K (e.g., Sinnhuber et al., 2011; Arnone et al., 2012; Manney et al., 2015a, b), sequestration in PSCs and denitrification led to depleted gas phase $HNO_3$ (Figure 4b) extending below 400 K in 2015/16. Sequestration in ice PSCs, and evidence of dehydration (in that low vortex $H_2O$

lingered well beyond the period with temperatures below the ice PSC threshold) is apparent in Figure 4c from about 420 K to above 550 K. Extensive chlorine activation is apparent from about 400 K up to above 600 K (Figure 4d,e), an upper extent comparable to that in the Antarctic. The upward tilt of ozone contours (Figure 4f) at levels from below 400 K to above 600 K beginning in early January indicates sufficient chemical ozone loss to exceed the replenishment by descent. This signature extends until mid-February at the higher levels, early March near 500 K, and continues into April at the lowest levels shown.



In the following, we focus on the evolution of the vortex and trace gas transport during the MFW on the individual isentropic surfaces marked by horizontal lines in Figure 4. 850 K (∼31 km, estimated from CAVE-ART vortex averaged altitude over the winter) is shown to represent the middle-stratospheric regime where the vortex decay is very rapid. 490 and 550 K (∼20 km and 22 km, respectively) represent the two regimes in the lower stratosphere with significantly differing vortex evolution during the

MFW. These lower stratospheric levels are near the maximum (490 K) and top (550 K) of the region of chemical processing. As seen in Figure 4, both of these lower stratospheric levels had exceptional chemical processing and large ozone loss by early March 2016.

## 4   The 2015/16 Major Final Warming: Vortex Breakup and Mixing

### 4.1   Overview of transport barrier and trace gas evolution

Timeseries of sPV gradients, $K_{eff}$, and MLS trace gases (Figures 5–7) as a function of EqL provide an overview of the vortex evolution throughout the 2015/16 winter. In the middle stratosphere at 850 K (Figure 5), sPV gradients and $K_{eff}$ indicate a consistently strong transport barrier along the vortex edge (strong maximum/minimum in sPV gradients/$K_{eff}$) through early March. The vortex area shrinks steadily through the winter, even as the vortex edge transport barrier strengthens and mixing outside the vortex increases (weaker sPV gradients, higher $K_{eff}$). This is consistent with the climatological development of the

Aleutian anticyclone, intensified mixing in the surf zone, decreasing vortex area, and accompanying strengthening of PV and tracer gradients along the vortex edge (e.g., McIntyre and Palmer, 1984; Leovy et al., 1985; Butchart and Remsberg, 1986; Harvey et al., 2002). Lower sPV gradients and higher $K_{eff}$ at midlatitudes in February indicate increasing activity in the surf zone (as has been previously reported, e.g., Haynes and Shuckburgh, 2000; Allen and Nakamura, 2002) consistent with the spreading of higher $H_2O$ values out from the vortex edge region. CO, because of its extremely strong gradients across the vortex

edge, provides a sensitive indicator of export of vortex air, and indicates periods of such enhanced transport in mid-February and early to mid-March. After the MFW began, the rate of vortex shrinkage accelerated rapidly, with the area enclosed within a transport barrier (sPV gradient maximum, $K_{eff}$ minimum) approaching zero by the end of March. The $H_2O$ and CO values show only slightly weakened gradients across the vortex edge in its final days, suggesting that most of the air in the remnants of the vortex was well confined within them until they disappeared.

In the lower stratosphere, at 490 K (Figure 6) the maximum PV gradients align closely with the minimum in $K_{eff}$ and indicate a strong barrier to mixing. The large and strong vortex persists until nearly mid-March, past the start date of the MFW. In early February, maximum $K_{eff}$ in mid-latitudes increases, suggesting that more vigorous mixing in the surf zone extends down into the lower stratosphere (consistent with the results of, e.g., Waugh and Randel, 1999; Harvey et al., 2002). Vortex area suddenly decreased and maximum sPV gradients/minimum $K_{eff}$ decreased/increased immediately after two small offspring split off the

vortex (around 13 March, second vertical red line), leaving the larger parent vortex even more distorted (see supplementary animation).

The signatures of mixing vary between trace gases depending on region and times because of differing horizontal gradients. Evidence of air from near the vortex edge mixing out into midlatitudes is seen in $N_2O$, $H_2O$, and $O_3$ during February in the





spreading of values previously characteristic of the vortex edge throughout the midlatitude surf zone. This is consistent with the common pattern of filaments of air being drawn off the vortex, around and into the anticyclone during this period. HCl shows consistent evidence of some mixing of very low values out of the vortex, with a concurrent extrusion of high ClO, in early February; these signatures are short-lived, since active chlorine transported out of the vortex in filaments rapidly decays

via both deactivation and mixing (e.g., Konopka et al., 2003; Tan et al., 1998; Marchand et al., 2004). Additional brief events of this sort are seen in mid-February and mid-March, and evidence of this mixing out of the vortex is also apparent in $HNO_3$ and $H_2O$. Small increases in vortex $N_2O$ and $O_3$ (just inside the overlaid sPV contours in the region of strong gradients) concurrent with the split suggest some mixing of extra-vortex air into the vortex region associated with that event, but the largest change following the split is the decrease in extra-vortex $N_2O$ and $O_3$ values, suggesting vortex erosion is the dominant process. Similar

evidence of increased mixing into midlatitudes after the vortex split is apparent in $H_2O$ and $HNO_3$.

At this level, minimum temperatures rose above the ice PSC threshold in late February, and the steady increase in $H_2O$ after that time indicates evaporation of ice PSCs. Minimum temperatures exceeded the chlorine activation threshold on about 13 March, nearly concurrent with splitting of the vortex (see Section 2.4). The evolution of HCl and ClO indicate rapid deactivation at this time, though non-zero ClO values lingered in the vortex until its disappearance in mid-April. Once chlorine

is largely deactivated (after mid-March in 2016), very high HCl values in the vortex make it a good tracer of transport (e.g., Manney et al., 2005), and substantial mixing into midlatitudes is apparent, consistent with the signature in the other species.

While the transport barriers seen in sPV gradients and $K_{eff}$ are weaker after mid-March, a significant maximum and minimum, respectively, remain along the edge of the rapidly shrinking vortex through early April. It is only at this time (apparent around 7 April in Figure 6) that very low $N_2O$ and $O_3$ previously confined to the vortex core are seen equatorward (in EqL) of

the strong PV gradients, indicating the final decay of the vortex.

A somewhat similar evolution is seen at 550 K (Figure 7), with a large vortex bounded by a strong transport barrier into early March, accompanied by increased mixing in midlatitudes in February consistent with filamentation and a more vigorous surf zone. In contrast to 490 K, however, while the vortex area shrinks after the onset of the MFW and vortex split, the maximum sPV gradients remained about as strong as before the split, and $K_{eff}$ continued to show a more pronounced minimum than

at 490 K. Consistently, $N_2O$ and $O_3$ (and other trace gases, not shown) do not suggest substantial mixing out of the vortex core until mid-April. Note that, similar to 490 K, vortex ozone was also strongly depleted at this level, resulting in very strong gradients along the inner edge of the vortex. At this level, however, the unperturbed morphology of ozone is such that vortex values are generally much lower than those outside the vortex prior to the onset of chemical loss.

The evolution of transport barriers and trace gases in the lower stratosphere is examined in more detail in Figure 8. Figure 8

compares the sPV gradients and $K_{eff}$ on each day during the period surrounding the MFW with the evolution of $M$, $N_2O$ gradients, and $O_3$ gradients as a function of EqL, as well as the evolution of $M$ as a function of "$M$-EqL" (see Section 2.4). The period covered is from 24 February, about 10 days before the beginning of the MFW, through 15 April, when the vortex was disappearing. The transport barriers shown by sPV gradient maxima, $K_{eff}$ and $M$ minima, and strongest trace gas gradients are all closely aligned. The transport barrier was near 65°EqL through the time of the vortex split, after which it shifted to about

75°EqL, indicating a substantial decrease in the total vortex area. The ozone gradients show a "dipole" pattern, with a large





positive extrema near $60°$EqL switching to a large negative one just poleward of it – this is the signature of increasing values in the outer vortex edge region changing to rapidly decreasing values moving into the vortex core where extensive chemical ozone loss has occurred. Note that on a given day the values of $M$ do not fall off as sharply on the high-EqL side of the peak as do those of sPV gradients and $K_{eff}$. This is primarily because $M$ values at EqLs higher than that of the vortex edge represent parcels that have, over the period of the calculation, been largely confined within the polar vortex, where they are likely to have spent some time near the edge in the region of high winds (thus travelling relatively long distances on average); conversely, the parcels at EqLs outside the vortex are largely in the surf zone where winds are weak and parcels do not linger near the vortex edge. The extrema of gradients in $N_2O$ and $O_3$ correspond well with those of the mixing diagnostics: The EqLs of the strongest negative $N_2O$ gradients closely matched those of the mixing diagnostics, while the largest negative gradients in $O_3$ were located around $5°$EqL poleward of that, consistent with their origin along the edge of the strongly ozone-depleted vortex core. The transport barrier presented by the vortex edge at this level rapidly weakened and moved poleward of $70°$EqL shortly after the split, and continued this progression through the end of March. In early April, a weak transport barrier was still apparent just equatorward of $80°$EqL, as reflected in the trace gas gradients. The location of the strongest $O_3$ gradient was aligned with the extrema in mixing diagnostics on this date, suggesting that the remaining vortex had largely been "stripped down" to its original core region. A similar pattern of evolution was seen at $550\,K$, and close alignment of mixing diagnostics and trace gas gradients was seen through the middle stratosphere (not shown).

Binning $M$ as a function of $M$-EqL is a convenient way of combining the size and strength of the polar vortex into a single diagnostic. Although $M$ is not a tracer (or tracer-like field), calculating EqL from any field provides an intuitive way of examining the area enclosed by its contours. We have found that plotting $M$ as a function of $M$-EqL is an easy way of showing the maximum distance traveled by a single parcel (at $90°$ $M$-EqL), which acts as a proxy for the strength of the vortex edge region over the 30 day trajectory period. If the vortex edge is strong and relatively wide, $M$ as a function of $M$-EqL will flatten towards the maximum value, indicating that a sizable fraction of parcels ended up "trapped" by the strong winds within the vortex edge and thus traveled long distances. While the slopes of the lines in late February and the first few days of March show this "flattened" shape above about 70 $M$-EqL, after the MFW starts, these move in toward 80 $M$-EqL, indicating that many fewer parcels ended up within the vortex edge region as the vortex weakened and decayed. Starting shortly after the vortex split, the flattened area virtually disappears, indicating a very small and/or weak transport barrier. Furthermore, the maximum $M$ value decreases on average by 1.51 Mm per day throughout the period, which shows the rapidity of the weakening of the vortex edge transport barrier. That the flattened area disappears while $M$ values still show a pronounced upward slope towards 90 $M$-EqL is consistent with the changes in the locations of extrema in the trace gas gradients and the picture of very small vortex areas lingering that were bounded by a significant transport barrier.

The EqL-based view presented above gives a global perspective on the evolution in vortex area and strength during the MFW period. This averaged view of mixing diagnostics shows that a small area of well-confined vortex air lingered through March, but by early April the transport barrier presented by the vortex edge was greatly weakened, and the potential for mixing was high. In the following, we focus on the synoptic evolution of the vortices and regional aspects of transport and mixing during the MFW period.



## 4.2 Synoptic evolution, vortex splitting, and local mixing

Figure 9 shows maps of MLS CO and $H_2O$ and $M$ in the middle stratosphere at 850 K, along with scatterplots of $M$ versus sPV for dates near the beginning of the MFW (7 March), near the time of the vortex split (13 March), as the offspring are shrinking (19 March), and just before the last offspring vortex disappears (4 April). In early March, the polar vortex at this

level was much smaller than earlier in winter (as seen in Figure 5), and was elongated and shifted far off the pole as is typical of a displacement SSW (Charlton and Polvani, 2007). A long filamentary tail was drawn off the main vortex and around the Aleutian anticyclone (whose "eye" can clearly be seen in $M$ just north of Alaska on 7 and 13 March, in the same region as anomalously low water). A single offspring eventually split off the parent vortex near mid-March, and both vortex regions quickly weakened thereafter. Strong confinement is indicated in the maps of CO and $H_2O$ throughout the period; even in early

April when the polar vortex has almost completely decayed, elevated CO and $H_2O$ signatures are seen in the remaining small vortex remnants. The maps of $M$ show features consistent with the trace gases and vortex edge region, as well as enhanced $M$ values in the anticyclone. The high $M$ values here indicate that the strong anticyclone acts as a transport barrier to trap air, and the spiral structure of the high values is consistent with tongues of air drawn off the vortex spiraling together with low latitude air forming persistent filamentary structures with very long transverse scales, as has been previously reported (e.g.,

Sutton et al., 1994). Consistent with the entrapment of air for many days at this time, ozone reached very low values within the anticyclone, forming a "low-ozone pocket" (e.g. Manney et al., 1995; Harvey et al., 2004) (not shown). The scatterplots of $M$ versus sPV initially show a "horseshoe" pattern that indicates the range of sPV values comprising the vortex edge region; that is, $M$ values show a relatively broad maximum in the vortex edge where parcels travel the furthest, though the picture is somewhat "blurred" by the high $M$ values associated with the anticyclone, which is associated with very low PV, but still

acts to coherently and rapidly transport air over long distances and thus gives enhanced values of $M$. The vertical red lines indicate the contour chosen to define the vortex edge in CAVE-ART, and show that this contour is near the maximum in $M$ as long as the horseshoe shape (indicating a significant transport barrier exists) is well defined; this indicates that the sPV contour used to define the vortex edge is well within the range of sPV values in the transport barrier (strong gradient) region. The vortex/vortices weakened very quickly at this level, and the horseshoe pattern rapidly disappeared. An animation of the

$M$ versus sPV scatterplots over the 24 February–15 April period is given in the supplementary information and shows this evolution in more detail.

Figures 10 and 11 show similar maps in the lower stratosphere, but with MLS $N_2O$ and $O_3$. At 490 K the vortex shrank rapidly between 7 and 13 March preceding a brief triple split after 13 March (nearly simultaneous with the split at 850 K, consistent with the barotropic structure of "split" SSWs, e.g., Matthewman et al., 2009). Of the three resulting vortices, the

largest and most stationary vortex over Siberia stayed the strongest and most coherent. The offspring over Greenland/Canada was also relatively strong as indicated by the low $N_2O$ and $O_3$ values on 13 and 19 March. In contrast, the offspring over Europe was much more transient and subject to significant mixing, indicated by higher $N_2O$ and $O_3$ values than in the parent and stronger offspring vortices. The maps of $M$ show qualitatively the same picture, with the largest $M$ values occurring in the edge region of the Siberian vortex. Large $M$ values also occurred in the edge regions of the smaller offspring vortices on



and shortly after 13 March, but decreased quickly thereafter as those vortices weakened and disappeared. Note that values of $M$ vary along the edge of a vortex in a manner that appears consistent with the trace gas gradients – for example, a region of very high $M$ values crossing near the pole north of Alaska on 7 March was associated with particularly strong gradients in $N_2O$ and $O_3$. Small differences in the shape of the vortex edge sPV contour and the maximum $M$ region reflect the fact

that $M$ is calculated from 30 days of information during a period of rapid change in the vortex. Scatterplots of $M$ versus sPV show strong horseshoe patterns that indicate the range of sPV values comprising the vortex edge region, and again indicate the appropriateness of the choice of vortex-edge sPV contour. As the vortex/vortices weakened, this horseshoe pattern became less well-defined, indicating the degradation of the transport barrier and increase in mixing; this process was slower in the lower stratosphere than in the middle stratosphere, consistent with the slower vortex break down. (Also see the animation of the $M$

versus sPV scatterplots given in the supplementary information.)

At 550 K (Figure 11) conditions were fairly similar, but the vortex split into only two offspring (according to our vortex-edge definition), with the Siberian vortex region initially larger and stronger. Both offspring shrank rapidly in this period, but the MLS maps show that air was comparably well confined in each at the same time. The smaller offspring (corresponding to the offspring over Greenland/Canada at 490 K) in particular was much stronger at 550 K than at 490 K. This is indicated by

15 the MLS trace gases showing significantly larger regions with trapped low $N_2O$ and $O_3$ depleted air all the way out through 4 April. This is also reflected in the maps of $M$, which show comparable values in the edge regions of both offspring into early April. Scatterplots of $M$ versus sPV also show horseshoe patterns that are similar to, but more pronounced than, those seen at 490 K. Even in early April as both offspring start to decay, a horseshoe pattern is apparent with a double "arch" structure showing the distinctly different strength of the two (both quite persistent) offspring; this double arch is apparent from 30 March

and through 7 April (see supplementary animation). As was the case at 850 and 490 K, the vortex-edge sPV contour lies near the maximum in $M$. Note, however, that there is a well-defined region of relatively high M seen in the map, and apparent in a local maximum in the horseshoe at lower sPV, on 13 March. There are also corresponding regions of low values in the MLS $N_2O$ and $O_3$ maps. Examination of daily maps (not shown) indicates that this was a coherent fragment of air from the vortex edge region that had sPV just below the threshold used by CAVE-ART, and this remnant persisted through about 18 March –

this represents the upward extension of the third offspring vortex seen at 490 K.

Figure 12 shows trajectory-based air parcel history maps at 850 K initialized on 12 March, just before the vortex split described above. On this date (column 2), a long filament of significant area had been drawn off the vortex and broken up into two pieces, each large enough to be identified in CAVE-ART as a vortex region. Though these were drawn off the vortex edge on 12 March, that air had been deep within the vortex 14 days before (column 1); this indicates that there was substantial mixing

within the vortex itself. After being drawn off, these narrow filaments quickly dissolved, with the air from them being widely dispersed through the hemisphere by 20 March (column 3) and even more randomized by 26 March (column 4). After splitting in two on 14 March, the air from the narrow elongated offspring vortex suffered a similar fate to those in these filaments and was quickly dispersed, while the other small offspring survived into early April (not shown). The rapidity of the dispersal of air shown here is consistent with the picture of the very rapid vortex break up in the middle stratosphere shown above in Figures 5

and 9.



Figure 13 shows similar air parcel history maps at 490 K during and after the vortex split. The air parcels in the vortices on 16 March (about two days after the split, row A) originated within the vortex 12 days earlier, with the parcels in the two small offspring vortices (green and blue) coming primarily from the narrower portion extending south near 30°E longitude. After the split, most of the air in the blue offspring vortex, which originated near 0° longitude, remained within a tight confined region for over two weeks, even after a vortex was no longer identified in that region. The 20 March initialization (row B, column 2), in which the blue vortex is the same one as the more persistent vortex region from the 16 March initialization, shows that this offspring vortex retained its identity into early April. The parent vortex (black) began to experience substantial filamentation in late March (row B, column 3). This main vortex was very small and had weakened by early April (row B, column 4), but was still identified as a vortex region on 10 April (row C, column 2) and maintained some coherence into late April (row C, columns 3 and 4).

At 550 K (Figure 14), the air parcels in the smaller (green) offspring vortex just after the split (row A) originated 12 days before primarily from a ring of air just inside the vortex edge. Most of the air in this offspring vortex remained coherent through March (row A, columns 3 and 4; row B, columns 2 and 3). Another small and very short-lived (about a day) offspring that broke off on 24 March (row B, column 2, blue vortex) was rapidly sheared out and the air originating in it wrapped around the outside of the parent (black) vortex on 8 April (row B, column 4). By this time the air in the green offspring vortex was starting to lose its coherence, though it can be seen (row C) that the air remained within a relatively confined region through 15 April (row C, column 3) and still showed some coherence on 20 April (row C, column 4).

Thus, compared to the rapid and wide dispersal of vortex air in the middle stratosphere, air from even small offspring vortices in the lower stratosphere maintained some coherence much longer after the vortex split in the lower stratosphere. At all levels, examination of the grey parcels – that is, all the parcels that were outside any vortex on the initialization day – without the overlaid vortex parcels indicates that few of them move into coherent vortex regions. That is, as long as the regions were large enough to be identified in CAVE-ART, they remained mostly devoid of air with extravortex origins. This indicates that the mixing during the vortex break up was largely one-way, with air mixing out of the vortices through filamentation as they eroded and lost their identity. This result is consistent with previous studies of dispersal of air from the lower stratospheric vortex (e.g., Chen et al., 1994; Manney et al., 1994), and with the picture of a shrinking and weakening vortex decaying primarily by erosion into midlatitudes.

Figure 15 summarizes how the transport and mixing processes described above affected trace gases in the lower stratospheric vortex. The top panels show the evolution of the vortex areas, and the MLS sampling of those vortices. An abrupt decrease in vortex area immediately followed the vortex split, with the total (sum of all vortices) area decreasing by about 40% and 30% at 490 and 550 K, respectively. This is consistent with the maps shown above and the time evolution shown in the supplementary vortex regions animation. At 490 K, the vortex size decreased more slowly, but steadily, thereafter, to about 3% of the hemisphere by the end of March, and less than 1% of the hemisphere by mid-April. At 550 K, the decay was more step-like, with another fairly rapid decrease in the area to a total of about 3% of the hemisphere in late March (near the time the second (blue) small, but very short-lived, offspring was pulled off and dispersed), followed by a sudden disappearance (that is, no vortex had area greater than about 0.5% of the hemisphere) by 12 April.





The MLS sampling of the large, strong vortex in January through mid-February included 500–700 measurements per day, but both its area and the number of measurements in it had dropped somewhat at all levels by 24 February (the start date of the panels in Figure 15). In general, the number of MLS measurements in the vortices closely tracks their area, and there are several MLS measurements in each: For every vortex region identified by CAVE-ART that lasted more than one day, the minimum number of MLS measurements on a day was at least six. This suggests that MLS usually provided relatively unbiased sampling of even small offspring vortices that were just larger than the $84°$EqL cutoff used by CAVE-ART. The number of MLS measurements begins dropping earlier, in the period between the beginning of the MFW and the split, because the vortex shifted farther off the pole to where MLS sampling is less dense. The rate of decrease in MLS measurements in the vortex at 550 K was steeper before the split than at 490 K, consistent with the vortex at that level being shifted farther out into midlatitudes. At 490 K, the steepest decrease in vortex MLS measurements was right around the split date. The minimum in number of MLS measurements shortly after the split (especially apparent at 490 K) is likely related to the fact that the vortex was shifted very far off the pole into midlatitudes and moved closer to the pole, into areas more densely sampled by MLS, in the following several days (see, e.g., Figures 10 and 11).

Vortex edge windspeeds show a deep minimum in the period between the start of the MFW and the split. Windspeeds showed some day-to-day variability after the split, but over all decreased steadily. The minimum just prior to the split arises largely because the vortex had already developed into two separate circulations that were only joined immediately prior to the vortex split by a narrow "bridge" with high PV but low windspeeds. As seen above, the offspring at 490 K were short-lived (about 5 and 7 days for the green and blue vortices, respectively), during which the windspeeds decreased rapidly along their edges. In fact, as seen in Figure 13, a coherent mass of air from the blue vortex persisted into April – represented in Figure 15 by the individual purple points labeled "transient", which mark the days on which the area of this region was larger than the $84°$EqL cutoff. The windspeeds around the edge of the parent vortex (black) remained stronger, though generally decreasing, into late April. A somewhat similar picture is seen at 550 K, with the windspeeds around the single offspring vortex (green) being weaker than those bounding the parent vortex, and the parent outliving the offspring; however, the offspring vortex at this level was much longer lived than those at 490 K. The evolution of vortex edge windspeeds is thus consistent with that of the transport barriers seen in the mixing diagnostics (sPV gradients, $K_{eff}$, and $M$) shown above.

The evolution of trace gases in the individual vortices is also consistent with the picture of mixing and vortex breakup seen above. At 490 K, $N_2O$ values were substantially higher in the blue offspring vortex, which persisted slightly longer than the green one, but was still rapidly sheared out into an elongated shape and weakened (as indicated by decreasing windspeeds). Examination of reverse domain filling (RDF, Sutton et al., 1994) maps initialized with MLS data (not shown) suggests that the rapid $N_2O$ increase in the blue vortex in the last two days may be an artifact of MLS not sampling the low values in the narrowest part of the vortex as it was sheared out. Figure 13 (e.g., row A, columns 3 and 4) shows rapid and widespread dispersal of the air from both blue and green offspring vortices, but with some of the air from the blue vortex remaining relatively coherent in a small region even after that vortex was no longer defined. $H_2O$ values were higher in the green offspring vortex because the air in that vortex came from nearer the edge of the parent vortex, rather than from the core where $H_2O$ was strongly depleted (Figure 13, row A, column 1). Average $H_2O$ in the blue offspring vortex was close to that in the parent (black) vortex, consistent





with that air coming from somewhat deeper in the parent vortex; this is also consistent with the appearance in Figure 6 of a "path" of low water crossing the vortex edge at the time of the split. Ozone was higher in both the green and blue offspring than in the parent because the air originated in the high $O_3$ collar near the vortex edge. It was highest in the green vortex because that air came from farther out towards the region of the $O_3$ maximum (see Figure 6). As was the case for $N_2O$, the increase in

the blue vortex in the last few days may be exaggerated by MLS sampling "missing" a narrow filament of vortex air.

At 550 K, $N_2O$ values were consistently higher in the single (green) offspring vortex than in the parent, indicating more extravortex or vortex edge air than in the parent, as shown in Figure 14 (row A, column 1). That air, however, remained largely confined within that vortex after the split (Figure 14, row A, columns 3 and 4), consistent with relatively constant $N_2O$ mixing ratios, and suggesting little additional mixing. RDF maps (not shown) at this level do not show obvious evidence of MLS

measurements missing filaments of vortex air. There was much less dehydration than at 490 K (see, e.g., Figure 4), so vortex values carried into the green offspring vortex were substantially higher than extravortex values, and the anticorrelation seen between $N_2O$ and $H_2O$ in that offspring vortex is consistent with this morphology. Low ozone values extended out to the vortex edge at 550 K (e.g., Figure 7), and thus the offspring carried very low ozone values with it. This offspring vortex was long-lived, and, though it shrank to an area too small to be cataloged a few days sooner than the parent, the air within both it

and the parent remained coherent into late April (Figure 14, row C, columns 3 and 4). Higher ozone air was drawn up around the parent vortex later on (e.g., Figures 11, and 14, row B, columns 3 and 4), consistent with the offspring vortex retaining lower ozone.

Examination of similar vortex averages of the shorter-lived species, HCl, ClO, and $HNO_3$ (not shown), indicates that the values of those species remained very nearly the same in all offspring vortices. Furthermore, RDF maps indicate that the range

of values in the small offspring vortices remained very close to those in the initialization fields 12 days earlier. This provides further evidence that the air in the offspring vortices was confined by an effective transport barrier as long at those vortices remained coherent. Thus, except in the period immediately surrounding the split, rapidly decreasing ClO and increasing HCl in all offspring resulted primarily from photochemical deactivation. While non-zero, albeit small, values of ClO (e.g., Figure 2) are apparent in the vortex averages through March, the area in which additional chemical loss could occur was small, less than

4% and 2% of the hemisphere at 490 K and 550 K, respectively.

## 5   Summary and Conclusions

We have analyzed meteorological fields from the MERRA-2 reanalysis and trace gas data from the Aura Microwave Limb Sounder (MLS) to provide an overview of the exceptionally cold 2015/16 winter and a detailed description of the the vortex breakup in a major final SSW ("major final warming" or MFW) that prevented chemical ozone loss from reaching record high

values. Our analyses utilized several mixing diagnostics, as well as a new package (CAVE-ART) for characterizing multiple vortex regions.

The 2015/16 Arctic winter was the coldest on record in December through early February. Lower stratospheric temperatures were at or near a record lows from late December into early February, and far below average from December through mid-



March. A substantial region of temperatures below the ice PSC threshold was present continuously from late December through early February, far longer than during any previously observed Arctic winter: The winter mean volume of air below the ice PSC threshold was over twice that previously seen. The chemical ozone loss potential, measured by the commonly used metric of volume of air below the chlorine activation threshold, was nearly identical to that in 2010/11 (when unprecedented Arctic

ozone loss occurred). The evolution of trace gases from MLS is consistent with the exceptional meteorological conditions: Vortex-wide dehydration was present between about 410 K and 520 K potential temperature, something never before observed in the Arctic. Denitrification was also exceptional, and extensive chlorine activation and chemical ozone loss began earlier than in all but one previous winter.

That lower stratospheric ozone loss did not reach values comparable to those in spring 2011 was primarily due to the

occurrence of an MFW beginning in early March 2016. This event had two critical consequences: First, while the the total volume of cold air during the winter was similar to that in 2010/11, that cold period ended significantly earlier in the winter in 2016, when ozone loss was slower due to less sunlight exposure. Second, the sudden vortex breakup in the MFW resulted in rapid dispersal of chemically processed air from the vortex and consequently curtailed chemical processing, which might have lingered for some time if chlorine had remained confined in relatively large intact vortex and thus deactivated more gradually.

The Arctic winter meteorology in 2015/16 was so remarkable that extensive study of numerous processes will be needed to fully characterize its consequences. In this paper we focus on one aspect of this exceptional winter: a detailed description of the event that limited ozone loss to an amount that, while larger than typical in the Arctic, was not unprecedented – the MFW and vortex breakup in early March. The MFW itself was an unusual SSW: The major SSW criteria were fulfilled when the vortex was a single elongated entity displaced far off the pole (a typical "displacement" SSW as defined by Charlton and Polvani,

2007), but a few days later the vortex split over a wide range of altitudes covering most of the stratosphere (behavior typical of a "split" SSW, e.g., Matthewman et al., 2009). Moreover, in a narrow range of levels in the lower stratosphere near 450 K to 550 K, that split was into three pieces.

In the middle stratosphere (exemplified herein by 850 K), mixing diagnostics and MLS trace gases show that by the time of the MFW the vortex had already shrunk and a strong Aleutian anticyclone and vigorous surf zone formed, consistent with

climatology. In mid-March, about a week after the MFW began, the vortex split into two very unequal pieces; the larger one rapidly sheared out and dispersed, while a very small coherent remnant of the other remained intact with relatively little mixing into early April. The evolution of MLS CO and $H_2O$ in the decaying vortices indicates that air within them remained well confined as long as they were intact. Snapshots of the function $M$ show a picture consistent with the the trace gas evolution, in that the vortex transport barrier decayed rapidly after the MFW onset.

The breakup of and dispersal of air from the vortex in the lower stratosphere was slower and more episodic, with largest changes in the short period surrounding the vortex split. Some of the specific consequences of the lower stratospheric vortex evolution (shown here at 490 and 550 K) during the MFW for transport, mixing, and dispersal of chemically processed air are as follows:

– At 490 K, two small offspring split off the main vortex in mid-March, but persisted for only about a week.





- At 550 K, the vortex split into two pieces, both of which remained well defined for over a month after the split.

- Mixing increased only slightly after the onset of the MFW around 7 March, but extensive mixing occurred in the few days during and after the vortex split in mid-March.

- Immediately following the split the total vortex area decreased by 30% to 40%, with the largest offspring covering about 4% of the hemisphere, and smaller offspring an additional 1 to 2% of the hemisphere.

- Following this period of intensive vortex erosion and mixing, air remained well-confined within the remaining offspring vortices.

- Abundances of MLS $N_2O$ and $O_3$ in the offspring vortices at 550 K remained closer to those in the parent vortex than at 490 K, indicating less mixing; this is consistent with the stronger transport barrier after the vortex split seen in the mixing diagnostics at that level, and with the greater persistence of the offspring vortices.

- ClO rapidly decayed in the offspring vortices as a result of a combination of rapid deactivation and dispersal of vortex air during the split.

- The evolution of ozone in the offspring vortices was dependent on the region within the parent vortex where the air originated, such that the offspring at 490 K contained higher values characteristic of the collar of undepleted ozone along the vortex edge, whereas at 550 K, low ozone values extended farther out into the vortex edge region and the smaller, but stronger, offspring vortex carried lower ozone than the parent.

- The "function $M$," when binned as a function of EqL, evolved consistently with the bulk mixing diagnostics (sPV gradients and effective diffusivity), but also revealed local variations (including relative strength of the offspring vortices, variations in the transport barrier around the vortex edge, and the dissolution of the individual vortices) that are consistent with the synoptic evolution of MLS trace gases.

In both the lower and middle stratosphere the mixing following the MFW was primarily via erosion and filamentation of the vortices as long as they remained intact. This resulted in wide dispersal and rapid mixing of air formerly in the vortex, but little extra-vortex air intruding into the vortex regions while they remained well-defined.

The major final SSW in early March 2016 was a remarkable finale to an already exceptional Arctic winter. The results presented here suggest the need for many further studies to assess not only how well the evolution of the vortex and trace gases throughout the 2015/16 winter fits with our current understanding of and ability to model lower stratospheric polar chemical processes, but also provides a unique addition to the already wide variety of natural experiments conducted via the immense variability in Arctic polar vortex evolution, longevity, and breakup. This new information is important for improving our detailed understanding of variations in dispersal of ozone depleted and/or chemically activated air from the vortex and its implications for present and future global ozone distributions. Further studies will include detailed analyses using similar methods to this work comparing the vortex breakup in 2016 with that in other winters, both Arctic and Antarctic. This is





particularly interesting given reported differences between years with early and late Arctic final warmings, which have not, in general, accounted for the suddenness of those final warmings (e.g. Waugh and Rong, 2002; Akiyoshi and Zhou, 2007); the 2011 vortex breakup, for example, was very late, but also quite sudden, whereas late final warmings in 2007 and 2008 were more gradual. In contrast to the Arctic, chlorine is typically deactivated well before the Antarctic vortex breakup (e.g.,

Manney et al., 2005; Santee et al., 2008), but the details and timing of that breakup still have important consequences – not only for local ozone minima over populated areas, but also for dilution of midlatitude ozone (e.g., Ajtić et al., 2004) and for radiative impacts of the Antarctic ozone hole (e.g., Polvani et al., 2011; WMO, 2014). Additional Lagrangian transport and air mass history studies, combined with analyses of Aura data over it's (so far) dozen year mission, will help quantify the fate of activated and ozone depleted air as the polar vortices decay.

In light of the 2012/13 winter, when an exceptionally strong vortex-split SSW resulted in record early winter ozone loss, and the 2014/15 winter, when a very brief, minor SSW resulted in record high vortex ozone values, the importance of the early and abrupt major final SSW in limiting ozone loss in spring 2016 once again emphasizes the complexity of the interactions between these extreme dynamical events and chemical processes in the stratospheric polar vortex. In each of these winters, the SSW events had dramatic consequences that were largely unanticipated. SSW characteristics are also expected to evolve with

the changing climate (e.g., Charlton-Perez et al., 2008; McLandress and Shepherd, 2009). We should thus expect the Arctic wintertime meteorology, and its impact on chemical processing, to continue to surprise us in the future, making continued comprehensive monitoring of stratospheric composition a critical priority.

*Acknowledgements.* We thank members of the MLS team at JPL for data processing/analysis, data management, and computational support (especially Luis Millan, Ryan Fuller, and Brian Knosp), for producing/providing the MLS dataset, and for helpful discussions (especially

Nathaniel Livesey, Michelle Santee, and Michael Schwartz). We also thank Ken Minschwaner for helpful discussions and comments on the manuscript. MERRA-2 reanalysis data were provided by NASA's GMAO, led by Steven Pawson, and we especially thank Kris Wargan for his helpful comments and assistance with usage of those. GLM's work on this paper was funded by and conducted under a contract from the Aura MLS Project at the Jet Propulsion Laboratory. Both authors contributed equally to this paper.

The datasets used are publicly available, the MLS data from

http://disc.sci.gsfc.nasa.gov/Aura/data-holdings/MLS/index.shtml,

corresponding DMP files from http://mls.jpl.nasa.gov,

and the GMAO reanalysis data from

http://disc.sci.gsfc.nasa.gov/daac-bin/DataHoldings.pl.



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





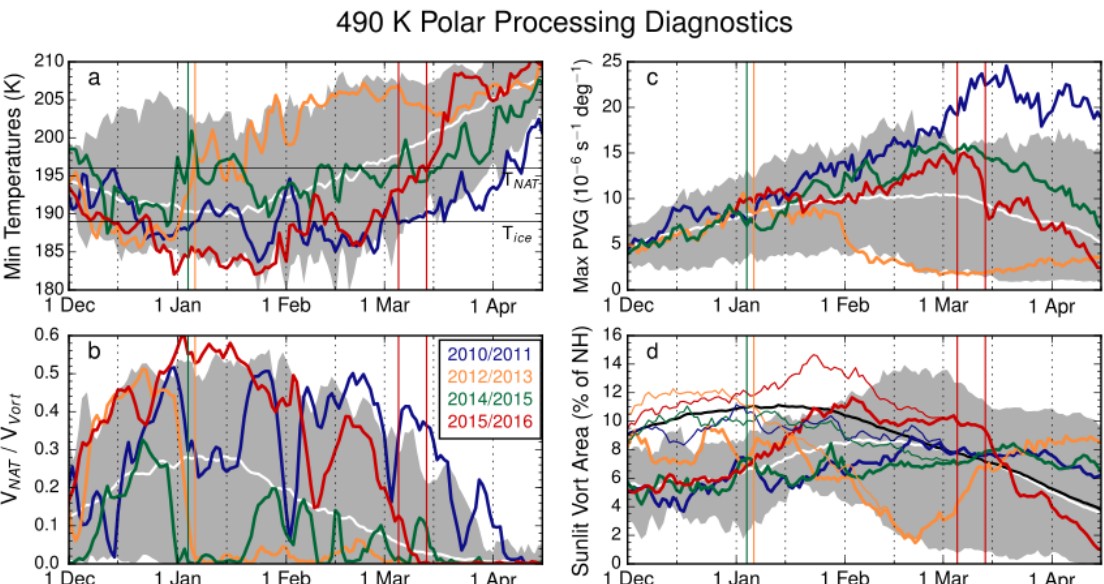

**Figure 1.** Time series of (a) minimum temperatures, (b) $V_{NAT}/V_{Vort}$, (c) maximum gradients of scaled potential vorticity as a function of EqL, and (d) sunlit area of the polar vortex from MERRA-2 in the 2010/11 (blue), 2012/13 (orange), 2014/15 (green) and 2015/16 (red) Arctic winters compared with the mean (white) and range (grey shading) of other years on record beginning with 1979/80. Thin vertical lines indicate significant SSW dates: 6 January 2013 and 5 March 2016 (first red line) are the dates when major SSW criteria were met in those winters; 4 January is the date when the 2015 minor SSW briefly split the vortex; 13 March (second red line) is the approximate date of the 2016 vortex split. In (a), the thin black horizontal lines indicate the NAT and ice PSC thresholds. In (d), in addition to the thick colored lines indicating sunlit vortex area, the thin colored lines indicate the total vortex area, with the thick black line showing the average total vortex area; at times when the thick and thin lines coincide, the vortex is fully in sunlight. (a), (c), and (d) are from data at the 490 K potential temperature level; (b) includes data integrated from levels between 390 and 580 K.





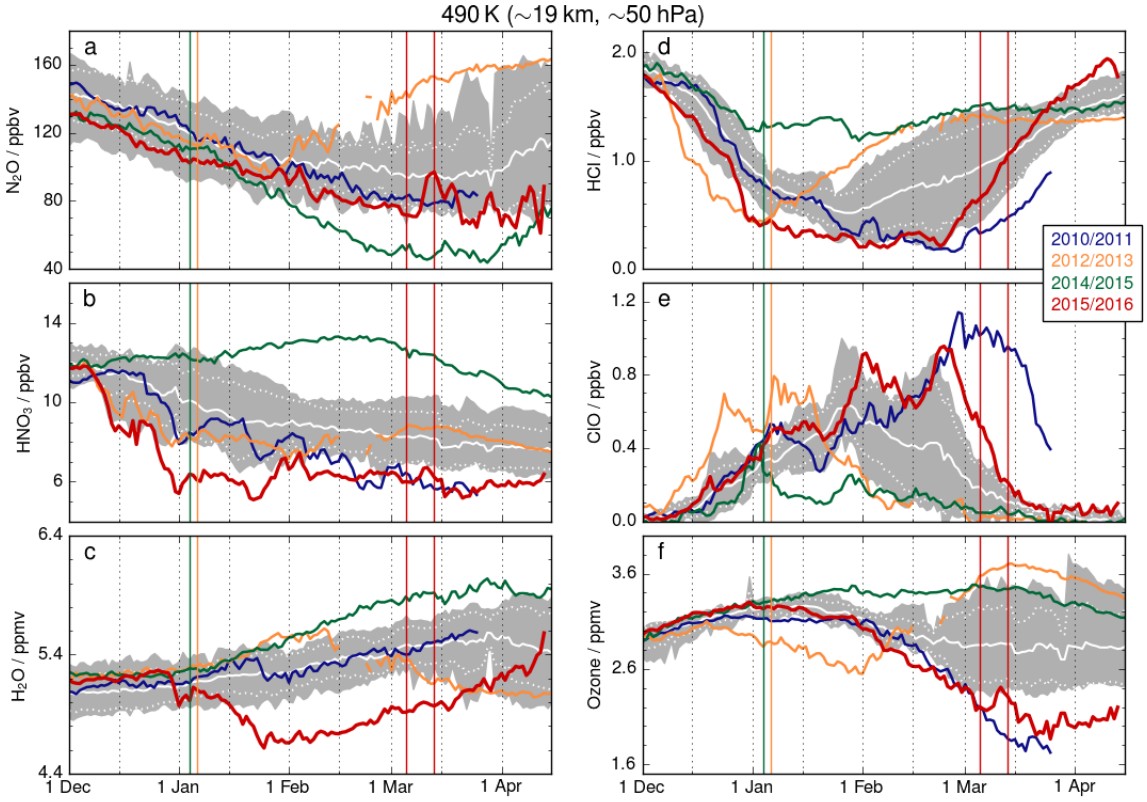

**Figure 2.** Time series of 490 K MLS vortex averaged $N_2O$ (a), $HNO_3$ (b), $H_2O$ (c), HCl (d), ClO (e), and ozone (f) in the 2010/11 (blue), 2012/13 (orange), 2014/15 (green) and 2015/16 (red) Arctic winters compared with the mean (solid white), one standard deviation range (dotted white) and minimum/maximum range (grey shading) of other years in the Aura record (beginning with 2004/05). Vertical colored lines are as in Figure 1. These vortex averages are calculated using the sum of the vortex regions identified by CAVE-ART with an 82° EqL cutoff to exclude un-climatological features due to very small vortex regions with anomalous characteristics in some years (the gap in the 2013 line shows a period when the lower stratospheric vortex was undefined by this criterion).





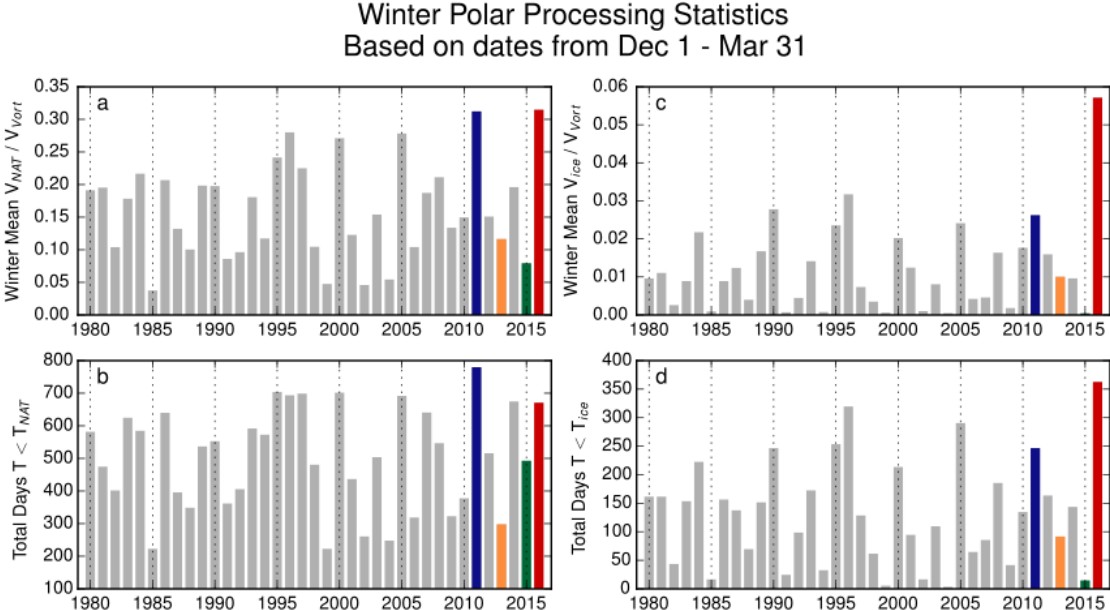

**Figure 3.** Winter polar processing statistics based on temperatures from the MERRA-2 reanalysis: (a) winter mean $V_{NAT}/V_{Vort}$, (b) number of days below $T_{NAT}$ summed over isentropic levels from 390 to 550 K, (c) winter mean $V_{ice}/V_{Vort}$, and (d) number of days below $T_{ice}$ summed over the same levels as (b). All bars are calculated from time series data limited to 1 Dec to 31 Mar. All year numbers are for the January of each winter; 2011, 2013, and 2015, and 2016 are highlighted as the blue, orange, green, and red bars, respectively.



**Figure 4.** Potential temperature/time series of vortex averaged (the sum of all regions identified by CAVE-ART using the 84°EqL threshold) MLS trace gases during the 2015/16 winter showing $N_2O$ (a), $HNO_3$ (b), $H_2O$ (c), HCl (d), ClO (e), and ozone (f). Horizontal black lines indicate 490, 550, and 850 K, the primary levels we focus on in this paper. The two black vertical lines indicate the SSW onset, and the ensuing vortex split. Version 4 MLS $N_2O$ at 100 hPa shows unphysical biases (Livesey et al., 2015a); $N_2O$ values below 430 K, where 100 hPa starts to significantly influence the vortex average, are thus not shown.





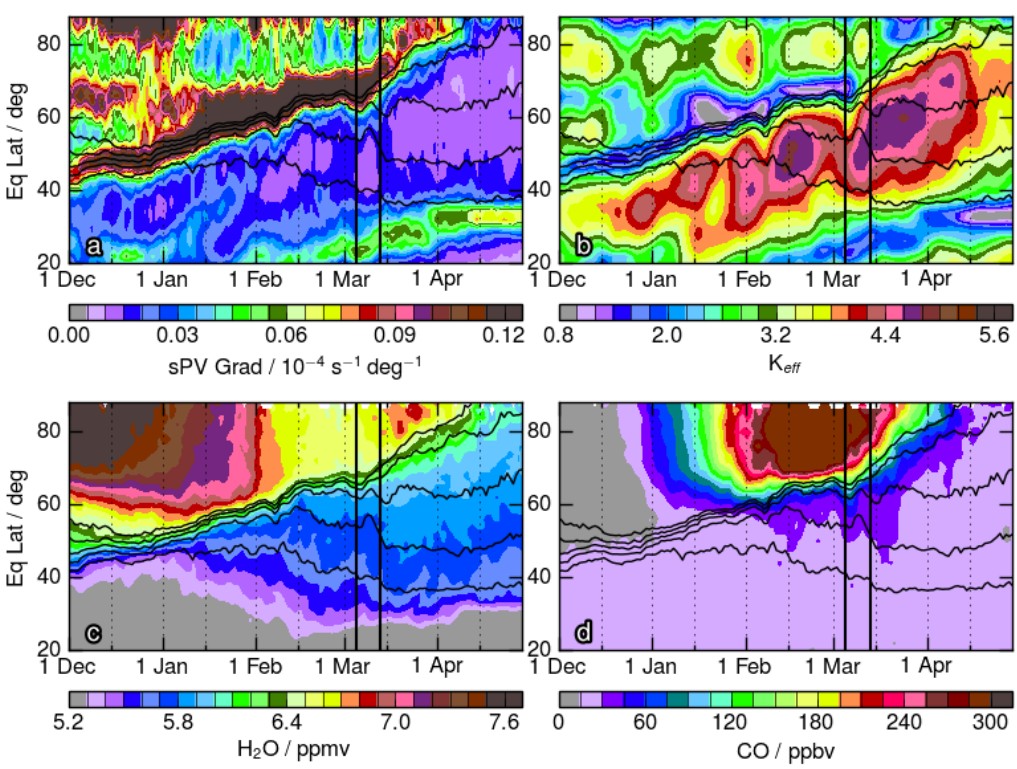

**Figure 5.** Equivalent latitude/time series at 850 K for 2015/16 showing MERRA-2 (a) sPV gradients and (b) effective diffusivity ($K_{eff}$), as well as MLS (c) $H_2O$ and (d) CO. Black contours show sPV values of 1.0, 1.2, 1.4, 1.6, and $1.8 \times 10^{-4}$ s$^{-1}$ in the vortex edge region. The vertical black lines indicate the onset day of the MFW and the following vortex split.





**Figure 6.** Equivalent latitude/time series at 490 K for 2015/16; as in 5, but showing (a) sPV gradients, (b) $K_{eff}$, (c) $N_2O$, (d) HCl, (e) $HNO_3$, (f) ClO, (g) $H_2O$, and (h) ozone.





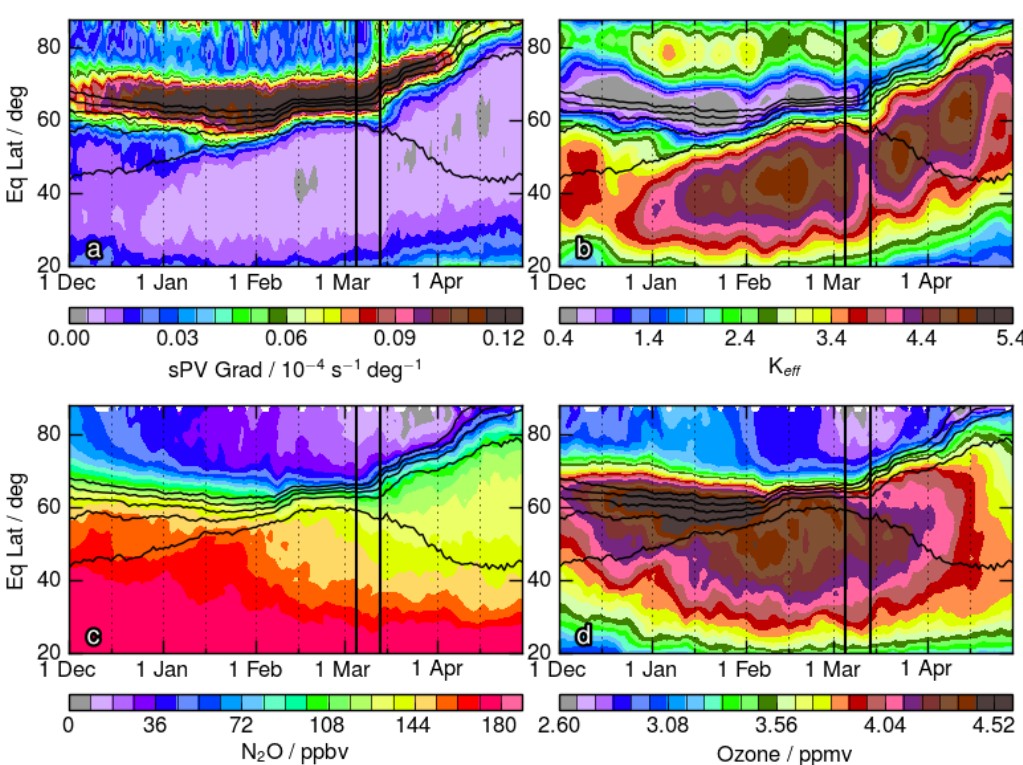

**Figure 7.** Equivalent latitude/time series at 550 K for 2015/16 as in Figure 6, but showing only sPV gradients (a), $K_{eff}$ (b), $N_2O$ (c), and ozone (d).



**490 K Mixing Diagnostics**

(colorbar: 24 Feb — 7 Mar — 20 Mar — 2 Apr — 15 Apr)

**Figure 8.** Equivalent latitude line plots of indicators of mixing and transport barriers at 490 K showing individual dates from 24 Feb through 15 Apr (see colorbar). The panels show (a) sPV gradients, (b) $K_{eff}$, (c) the function $M$, (d) EqL gradients in $N_2O$, (e) EqL gradients in ozone, and (f) $M$ as a function of $M$-based equivalent latitude ($M$-EqL). Quantities in panels (a) through (e) are all functions of PV-based EqL; panel (f) is the only exception. Note that the units of $M$ are in megameters (Mm, $10^6$ m).





**Figure 9.** Orthographic maps of 850 K MLS CO (first row) and $H_2O$ (second row), and function $M$ from MERRA-2 (third row), along with scatterplots of $M$ versus scaled potential vorticity (fourth row), for individual dates during the major final warming (columns are 7 Mar, 13 Mar, 19 Mar, and 4 Apr, respectively). The white contours in the first three rows, and red lines in the bottom row show the sPV value used in CAVE-ART to define the vortex edge at this level. Note that the units of $M$ are in megameters (Mm, $10^6$ m).







**Figure 10.** As in Figures 9, but at 490 K, and showing MLS N$_2$O and O$_3$ (in first and second rows, respectively).




**Figure 11.** As in Figure 10, but at 550 K. Note that contour ranges are different than at 490 K.





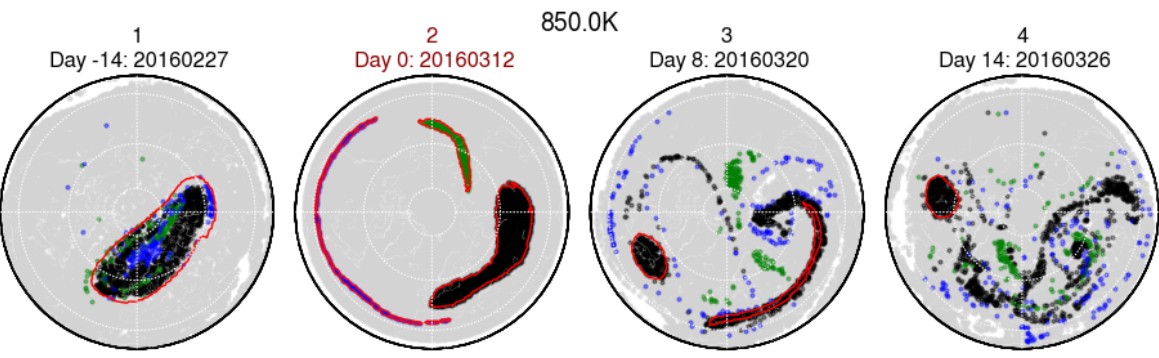

**Figure 12.** Trajectory-based parcel history maps at 850 K showing the locations of air parcels initialized inside vortex regions as defined by CAVE-ART on 12 March. Parcels are colored green, blue, or black if they were inside a valid vortex region on the initialization date (column 2, red labeling); otherwise the parcels are colored grey. Columns 1, 3, and 4 show the locations of these parcels 14 days before, and 8 and 14 days after initialization, respectively. The red contours show the vortex regions identified by CAVE-ART in MERRA-2 data (subsampled to match the $1.25° \times 1.0°$ longitude latitude grid used by the trajectory runs) on each date.







**Figure 13.** As in Figure 12, but for 490 K and showing 12 rather than 14 days back. Initialization dates (column 2) are 16 and 20 March and 10 April.







**Figure 14.** As in Figure 13, but for 550 K. Initialization dates (column 2) are 16 and 24 March and 6 April.





**Figure 15.** Vortex characteristics and MLS trace gas averages in individual vortex regions in the lower stratosphere. Top panels show the area of each vortex (shading) along with the number of MLS measurement points inside each vortex on each day (lines/symbols). Second row shows windspeeds from MERRA-2 averaged around the edge of each vortex. Succeeding rows show averages of MLS $N_2O$ (third row), $H_2O$ (fourth row), and ozone (fifth row) in each vortex region, with shading indicating the standard deviation. In the MLS averages, the dark grey lines indicate the "bulk" values (that is, within the vortex edge contour even if the area does not exceed the 84°EqL cutoff, see Section 2.4).