# Peer review of "The major stratospheric final warming in 2016: Dispersal of vortex air and termination of Arctic chemical ozone loss"

_Atmospheric Chemistry and Physics, 2016_

## Short Comment (SC1)

**The Method of Lagrangian Descriptors**

Alfonso Ruiz-Herrera[*]

September 11, 2016

The $M$-function was used for the first time in the pioneering work

- A.J. Jimenez Madrid and A.M. Mancho, *Distinguished trajectories in time dependent vector fields,* **Chaos 19** (2009), 013111.

In two papers published in the same journal,

- A. Ruiz-Herrera, *Some examples related to the method of Lagrangian Descriptors,* **Chaos 25** (2015), 063112,

- A. Ruiz-Herrera, *Performance of Lagrangian Descriptors and Their Variants in Incompressible Flows*, **Chaos** (in press),

I have shown that the contours of the $M$-function have no implications in the detection of barriers to transport. Mancho, Wiggins, and their co-workers have posted on *arxiv* (without external peer review process) a paper that criticizes my work, namely

- F. Balibrea-Iniesta, J. Curbelo, V.J. Garcia-Garrido, C. Lopesino, A.M. Mancho, C. Mendoza, and S. Wiggins, *Response to: "Limitations of the Method of Lagrangian Descriptors"*[arXiv: 1510.04838]. arXiv preprint arXiv:1602.04243. (2016).

To defend their tool, they simply say that the method of Lagrangian Descriptors (LDs) does not involve the contours of the $M$-function. It is easy to observe that they misrepresent what they have actually done, (see comments below). We mention that Manney and Lawrence have detected the barriers to transport from the contour-lines of the $M$ function.

*Chaos: An Interdisciplinary Journal of Nonlinear Science* offers the possibility of comments to regular papers, see *http://scitation.aip.org/content/aip/journal/chaos*. In fact, Mancho, Wiggins and their co-workers submitted a reply to *(Ruiz-Herrera 2015)* the last year. I strongly encourage to submit their new critique again in order to avoid misleading conclusions in the literature. (Limitation of the Method of Lagrangian Descriptors was the preliminary version of *(Ruiz-Herrera in press)*).
* * *
[*]Departamento de Matemáticas, Universidad de Oviedo, Spain (alfonsoruiz@dma.uvigo.es, ruizalfonso@uniovi.es).

**1 Comments on "Lagrangian Descriptors for Two Dimensional, Area Preserving Autonomous and Nonautonomous Maps, Communications in Nonlinear Science and Numerical Simulation 27 (2015), 40–51 by C. Lopesino, F. Balibrea, S. Wiggins, A.M. Mancho"**

In the aforementioned paper, Lopesino *et al.* have provided mathematical theorems to support the performance of the method of Lagrangian Descriptors in discrete systems.

Specifically, given $\{x_n, y_n\}_{n=-N}^{n=N}$ with $N \in \mathbb{N}$ an orbit of length $2N + 1$ generated by a two-dimensional map, we consider

$$MD_p(x_0, y_0) = \sum_{i=-N}^{N-1} |x_{i+1} - x_i|^p + |y_{i+1} - y_i|^p \quad \text{with } 0 < p < 1. \tag{1.1}$$

They stated the following result:
Let

$$\begin{cases} x_{n+1} = \lambda x_n \\ y_{n+1} = \frac{1}{\lambda} y_n \end{cases} \tag{1.2}$$

with $\lambda > 1$.

**Theorem 1.1.** *Consider a vertical line perpendicular to the unstable manifold of the origin. In particular consider an arbitrary point $x = \bar{x}$ and a line parallel to the $y$ axis passing through this point. Then the derivative of $MD_p$ with $p < 1$, along this line becomes unbounded on the unstable manifold of the origin.*

The theorems presented in *( Lopesino et al. 2015)* are a consequence of the diagnostic itself because $MD_p$ is non-smooth if, for some iteration,

$$x_{i+1} = x_i \ \ or \ \ y_{i+1} = y_i \tag{1.3}$$

Therefore, there are unbounded behaviours of the derivatives of $MD_p$ in those points independently of the dynamical behaviour of the system. A more detailed discussion can be found in *(Ruiz-Herrera in press)*. As emphasized in my work, Theorem 1.1 gives no specific recipe for detecting unstable manifolds in general systems, it only works for systems satisfying (1.3). We also discussed in *(Ruiz-Herrera in press)* that Theorem 1.1 does not provide a mechanism to approximate the invariant manifolds when $N \longrightarrow \infty$ (this property was mentioned in Section 2.1.2 in (*Lopesino et al. 2015*)).

**2 Critiques to the method of Lagrangian Descriptors in the literature**

Apart from my work, many authors have criticized the method of Lagrangian Descriptors. Next, I give some examples:

Most applications of the $M$ function have been published in *Nonlinear Processes in Geophysics.* A prerequisite for reliable predictions is the independence of the observer or objectivity. Unfortunately, as Haller nicely emphasized in

- *Non-objectivity of the M function and other thoughts*, Interactive discussion in Nonlinear Processes in Geophysics about the paper *Detecting and tracking eddies in oceanic flow fields by Rahel Vortmeyer-Kley, U. Grave, and U. Feudel*.

LDs are not objective. A.M. Mancho, as executive editor of that journal, provided a reply (without external evaluations) to Haller's comment. In her reply, she discussed the performance of the $M$-function in $x' = -y \; y' = x$. Of course, a concrete example is not sufficient to refute a general property. Moreover, she claimed: "the contours of $M$ are in 1-1 correspondence with the trajectories of the system". There are several contradictions in this claim. First, the mentioned 1-1 correspondence is an exceptional feature of this particular system. For instance, it does not hold in $x' = x, y' = -y$. Importantly, her claim contradicts the fact exposed in *(Ballibrea-Iniesta et al 2016)* because they indicated that method of Lagrangian Descriptors does not involve the contour-lines of $M$.

In

- Fabregat, A., Mezic, I., and Poje, A. C. (2016). Finite-time Partitions for Lagrangian Structure Identification in Gulf Stream Eddy Transport. arXiv preprint arXiv:1606.07382.

we can find in page 16 after definition 3:

"The approaches based on integral of positive functions along a trajectory can fail to uniquely identify underlying objects. Specifically, the notion of Lagrangian Descriptors is based on this, but this notion is neither new, as it was proposed much earlier, nor complete."

**3    Rebuttal to *( Balibrea-Iniesta et al. 2016)**

The method of LDs has been broadly applied by Mancho, Wiggins and their co-workers ($\sim 20$ papers). They always analyze the contour-lines of the $M$-function (typically from data). For this reason, the statement of my results was on the behaviour of the contours of the $M$-function. However, they stated in their arxiv submission:

*The term "singularity" does not refers to properties of the contour-lines of $M(x_0; t_0, \tau)$, but to points at which certain derivatives of $M(x_0; t_0, \tau)$ do not exist. While this is discussed somewhat in the references Mendoza and Mancho 2010 and Mancho et al 2013, it is made precise in Lopesino et al 2015*

**My response:**

- As emphasized above, the conclusions of the theorems in *( Lopesino et al 2015)* are a consequence of the mathematical definition of the diagnostic. Therefore, those results have no dynamical significance. On the other hand, one can find in the introduction of that paper (page 41, line 10):

  *Hyperbolic structures are revealed as* **singular features of the contours of the lagrangian descriptors***, but the sharpness of these singular features depends on the particular norm chosen.*

  The analysis of the derivatives is carried out to extract information in the geometry of the contours of LDs. As emphasized in any paper of Mancho, Wiggins and their co-workers, the detection of barriers to transport always involves the contour lines of $M$. For instance, see

figures 1-4 in *(Mendoza and Mancho 2010)*; figures 1-3, 5,7,9 in *(Camara et al 2013)* (the contour-lines of the derivative of $M$ are missing); see figures of section 4 (applications) in *(Lopesino et al 2015)*; see figures 1-5 in *(Garcia-Garrido et al 2015)* and so on. In fact, in their last application *(Garcia-Garrido et al 2015)*, Section 4.1 says:

*The structure of the M function shows, at low $\tau$ values, a smooth pattern such as that visible in Fig. 1a [...]. On the other hand, Fig 1.b (computed for $\tau = 20$ days) illustrate how the structures of M evolves for large $\tau$ towards less regular structures. By this we mean that sharp changes of M values occur in narrow gaps, forming filaments that highlight stable and unstable manifolds.*

- The mathematical foundation of the method of LDs in continuous dynamical systems is given in *(Mancho et al 2013)*. In that paper, one can find the term "contour" exactly 68 times and expressions as

  *The contours of $M_1$, for any fixed $\tau$, are smooth circles surrounding the origin.*

  *As already noted the patterns of the contours displayed in Fig. 1 depends on $\tau$. For $\tau$ small the contours are smooth but for increasing $\tau$ they develop singular features along the unstable and stable manifold.*

  *Singular contours of LDs correspond to invariant manifolds.*

- In *(Balibrea-Iniesta et al 2016)*, the authors have used two outputs to capture barriers to transport in their systems. This is a contradiction with:

  *The Lagrangian descriptors have the capability of revealing both the stable and unstable manifolds in the same calculation.*

  See Section 6 in *(Mancho et al 2013)*. This property has been emphasized several times in *(Mendoza and Mancho 2010)*.

- Figure 1 in *(Ruiz-Herrera in press)* is exactly the same as Figure 1 (c) and 2(a) in *(Mancho et al 2013)*. However, they introduced the following misleading comment in *(Balibrea-Iniesta 2016)*, (see caption in figure 1):
  *This figure should be compared with figure 1 of the comment of Ruiz-Herrera.*

- The unbounded behaviour of the partial derivatives of $M$ can not capture the dynamical behaviour of a dynamical system because $M$ is typically unbounded. A more detailed discussion can be found in *(Ruiz-Herrera in press)*.

---

## Referee Comment (RC1) · A. Dörnbrack (Referee) · 17 Aug 2016

Review of

The major stratospheric final warming in 2016: Dispersal of vortex air and termination of Arctic chemical ozone loss

by Gloria L. Manney and Zachary D. Lawrence

This paper is a well-written, comprehensive study of the fate of the northern hemispheric polar vortex in spring 2016. The paper is well-structured into an Introduction, a Data and Method section, overviews the 2015/2016 polar vortex evolution in Section 3, and, finally, focusses on the early vortex breakup in March 2016 and the subsequent

mixing of vortex air with mid-latitude air. Section 5 summarizes and concludes the paper with clear statements and with a friendly wink and hint to follow the NH vortex evolutions in the future with care. I'm sure, the authors will do so as they possess the suitable diagnostic tools to analyze global satellite and meteorological data in an efficient way.

The Arctic winter 2015/16 was extraordinarily cold and the vortex-wide temperatures felt as low that the conditions in January resembled those of the Antarctic in terms of chemical composition and PSC appearances. Besides the fascinating subject of the recent winter, it is the clarity in structure and writing which make the paper a joy to read.

The Introduction sets the scene by stating "that SSWs affect Arctic lower stratospheric chemical ozone loss in ways much more complex than a simple association of low (high) temperatures with more (less) ozone loss". So it is consequent to read the motivation as: "Thus, understanding the complex relationships between SSW dynamics, stratospheric vortex evolution, and chemical composition and processing, is critical to diagnosing and predicting ozone loss and recovery in the Arctic and its climate consequences." After a short historical view, the authors come back to the topic by discussing the interannual variability of NH winters (minor/major SSWs and dates of final warmings) in close relation to the chemical ozone loss. Some of the main results are already anticipated in the fourth paragraph: (1) "the 2015/16 Arctic winter was the coldest on record (since at least 1979)"; by the way, Matthias et al. (The extraordinarily strong and cold polar vortex in the early northern winter 2015/16, GRL, under review) showed that it is was indeed the coldest in the recent 68 years. (2) a major final warming "beginning in early March 2016 resulted in the breakup of and dispersal of chemically processed air from the vortex, which halted chemical loss much earlier than in 2011".

Section 2 reviews the data sources and the methods. It is impressive to see that the latest versions of MERRA-2 and MLS data are used. The diagnostic quantities and tools are presented systematically. Even newer developments which are in the pro-

cess of publishing are explained in a comprehensive way. The authors apply a broad spectrum of well-established and newly developed diagnostics to quantify the spatio-temporal variation of the various trace gases and dynamical quantities specifying the mixing in the surf zone of the polar vortex.

The Overview of the 2015/16 vortex evolution and composition focusses on thermal and chemical aspects. For a reader not so familiar with all the peculiarities of the previous winters less direct comparison would be advantageous. Maybe, sentences like (around line 320) "Ozone continued to decrease in the vortex at a rate slightly faster than that in 2011 until the beginning of March 2016. If uninterrupted, ozone values would have been expected to drop lower than those in 2011 by mid-March." could be slightly reformulated to give for example explicit values of rates. But this, for sure, is only a matter of taste. At the end, main results for the winter 2015/16 (in relation to previous ones) are presented and key words are: "leading to unanticipated extremes in Arctic polar processing, the 2015/16 winter stands out as yet another unexpected extreme in variability of the Arctic winter stratosphere." , "The period of over a month, from late December through early February, with temperatures below the ice PSC threshold was unprecedented in the Arctic", "much greater degree of dehydration", and "extreme denitrification", "extensive early winter chlorine activation", and "chemical ozone loss began early". And finally, we read: "Thus, the critical factor resulting in less ozone loss than in 2011 was the much earlier increase in temperatures and vortex breakup in 2016."

The Section 4 about the 2015/16 major final warming and the resulting vortex breakup and mixing exemplifies the trace gas evolution and the strength of the transport barrier, and the mixing especially in the surf zone by refined analyses at different isentropic surfaces representing the conditions in the middle and lower stratosphere. I'm not the expert to evaluate the details of the applied diagnostics but the text reads logical and the conclusions are based on well-funded results from the respective simulations.

Altogether, the paper can be published in the present form!

**[ACPD](https://www.atmospheric-chemistry-and-physics.net/)**

Interactive
comment

---

## Referee Comment (RC2) · Anonymous Referee #3 · 9 Sep 2016

This paper is certainly comprehensive, appropriate for ACP and most probably, correct. It was, however, also difficult to read and review. It comes across, at least to this reviewer, as a so-called "core dump" of information. As a consequence, I readily admit that this review is probably incomplete and that I probably missed pieces of information that to the authors, at least, they would deem critical. My comments are therefore (with one exception) more editorial than scientific.

General 1. One general science question that I think could use a greater exposition is the question of the MFW. The authors imply that this hybrid event in 2016 is unusual. Some context as to its occurrence frequency would be helpful. Is this the first to occur in the AURA record?

2. As far as presentation, Section 3 is a case in point as exemplifying my concerns. Figures 1-3 are introduced in rather random order, with lots of information, which, while not technically wrong, may well be irrelevant. The authors present a whole bunch of figures (14 panels in all for Figures 1-3) and then jump back and forth in a scatter shot discussion. This is very taxing to read. The first three paragraphs do not even discuss the 2015-16 season, but rather present a literature review of 3 three previous winters. Line 27 on page 7 is a good example. The statement is simple- temperatures in a particular year (a year which was not the subject of the present paper) were cold enough to activate chlorine for a prolonged period of time. So why do we need to refer to four separate figure panels (Figure 1a and b Figure 2d and e?) to make this simple point (which again is irrelevant to the subject of this paper that is nominally about 2016)? In fact, I don't understand why Figure 2 is referred to here. Is it because ClO was going up? That is not explained.

3. Adding up all the panels in 15 figures, the paper contains 128 separate graphs. I confess that I found it difficult to subject each and every one to the scrutiny they probably deserve; I do nonetheless strongly suspect that they are not all necessary. As an example, I did examine one specific panel- that of Figure 1b. All references to Figure 1b occur with a simultaneous reference to Figure 1a. I therefore conclude that Figure 1b can't be necessary since it never is referred to independently of Figure 1a. So it should be deleted. Especially since they never describe it (what is V_nat/V_vort?- they briefly mention it on page 9, but not in the context of Figure 1).

4. I also think Figure 3 is unnecessary. Not that it's technically incorrect, but it adds no new information that is not conveyed in Figures 1 and 2. Indeed, their concluding sentence on lines 19-20 of page 9 can easily be gleamed from Figures 1-2.

Minor

1. Figures 5-14 (with the exception of Figure 8) are essentially 3 sets of three figures for 850, 490, and 550 potential temperature surfaces. It would be useful to have a few

introductory few sentences at the beginning of Section 4 explaining why they chose these three levels. Even if it was empirically determined that they were good representative levels, they should at least say that. As it reads, it just says (for example) Figures 5-7 without telling the reader where you are going with this. You have to read almost 2 pages of the draft before you find out that these 3 figures are for three separate altitudes.

2. Figure 6 vs Figure 7. If I understand correctly, the text on page 11 suggests (line 25, compared with line 7) that one difference is that N2O and O3 do not show mixing out of the vortex at 550K but they do at 490 K. Looking at Figures 6c and 7c, I see no difference. Am I supposed to?

3. Figure 1c: what is Max PVG? Those three letters do not appear anywhere else in the text or figure captions.

4. Figure 15: What do the colors mean? There is a label that says "first", "second" etc, but doesn't explain what those terms mean other than "bulk". Are they related to the colors of various fragments in Figure 14. If so, it should say so.

5. Abstract: Line 20. Where do they show chlorine in the offspring vortices? Figure 15 does not show chlorine. There are cryptic references to chlorine activation and deactivation scattered throughout the text, but I could not find where it pointed to a figure saying "this shows the deactivation of chlorine etc. etc."

---

## Referee Comment (RC3) · R. Thiéblemont (Referee) · 19 Sep 2016

This paper provides a thorough description of the evolution of the 2015/2016 Northern Hemisphere stratospheric winter until the breakup of the polar vortex and the dispersal of its fragments. This winter was unique: while it initially presented the characteristics for an unprecedented ozone loss (i.e. prolonged temperatures below ice polar stratospheric cloud thresholds), an anomalous early and strong major final warming interrupted the ozone depletion process. Dynamical and chemical processes are characterized using Microwave Limb Sounder satellite trace gas measurements, and advanced mixing and polar vortex diagnostics derived from meteorological reanalysis.

This case study of the winter 2015/2016, and its comparison with the series of singular

recent winters, very well illustrates the complexity of the dynamical and chemical interactions that drive Arctic ozone depletion. In my opinion, this paper is important as it further contributes to showing that each Arctic winter season is unique and that substantial research efforts are needed to better understand their extreme variability and the consequences of this variability (e.g. on ozone depletion, stratosphere/troposphere couplings, . . .). The methods and diagnostics used in this study are scientifically sound and relevant. The analysis is very carefully conducted. My main criticism rather concerns the form: the main text and its figures are extremely dense and contains a lot (too much?) of information so that it is sometime hard to differentiate what is important from what is more anecdotal. While in some places the degree of detail seems to me exaggerated (e.g. p10l32-p11l10 where tracer extrusions are discussed while not really obvious), in other places, including further details may help to make the paper easier to follow (see comments below). Hence in my opinion, this paper is suitable for publication in Atmospheric Chemistry and Physics after consideration of the specific (minor) comments and suggestions provided below.

Specific comments:

1) p3l32: Typo change "MERRRA" to "MERRA"

2) p4l20: Please provide further detail on the way the potential vorticity is scaled. The sPV is widely used throughout the paper so few precisions about it may be useful for the readers.

3) Diagnostics (i.e. sections 2.3 & 2.4): This paper makes use of a very high number of diagnostics to describe mixing processes, transport, vortex size and so on. Although the different diagnostics are very well explained in the main text, non-expert reader may quickly be lost once the description of the (dense) analysis begins. The authors may consider adding a table which gives a summary describing (briefly) the different diagnostics and their usefulness.

4) p7l25: "The 2010/2011 winter . . .". Please mention the associated color line in

bracket to help the reader.

5) p8l1: "In early January 2013 . . .". Same here, please mention the associated color.

6) p8l2: ". . . strongest "vortex-split" SSWs on record. . ." What does strong mean here? What defines the strength of a SSW (persistence, temperature, vertical extension, . . .?)? Please clarify.

7) p8l6: "2014/2015. . .". Please mention the associated color.

8) p8l6: ". . . brief minor SSW . . .". Please give the date. (I guess early January)

9) p8l13: Typo change "though" to "through"

10) p8l14: "unprecedented". On MERRA record? Please clarify.

11) p8l31: ". . . in 2015/1016, 2012/2013, and 2010/2011 . . .". Why not 2014/2015? The green curve looks similar in early winter on Figure 2a.

12) p11l1-2: "This is consistent [. . .] anticyclone during this period". Does anticyclone refer to the Aleutian High here? Please clarify.

13) Figure 8: Please replace y-axis "Effective diffusivity" by "Keff" to be consistent with the main text (p11l30).

14) p11l33: ". . .Keff and M minima. . ." Is it not rather M maximum? M maximum ⇔ vortex edge ⇔ transport barrier.

15) Figure 9-14: Please make the continents more visible on maps and provide at least on longitude coordinate. Otherwise it is quite hard to follow Figures together with the main text and the geographical location that are refereed (e.g. Alaska p13l7 but also at other places).

16) P13l12: ". . . in the anticyclone.". Is it not ". . . in the edge of the anticyclone" that the M values are the strongest?

17) At 550 K, a doubled vortex edge appears in the main vortex fragment (see Fig 11,

14) from beginning of April. Is this an artifact or a real structure? Please comment on this.

18) Figure 13-15 (and associated text). The green and blue offsprings seem actually switched between the 490 and 550 K levels. If indeed this is the case, it may be confusing. Therefore, it may be more relevant to keep the same color for the upward extension of the same offspring.

19) p15l17-19: May this vortices coherence dependence with height be partly related to differences in diabatic processes with height?

20) p16l7: "... begins dropping earlier, ...": earlier than when? Please clarify.

21) p16l7: "... period between the beginning of the MFW and the split ...": is it the period between the two dashed lines? Please clarify.

22) p16l19-21: "In fact, as seen in Figure 13, a coherent mass of air from the blue vortex persisted into April – represented in Figure 15 by the individual purple points labeled "transient",...". I guess these transient vortices are those seen in the supplementary animation and labelled 4, 5, 6, 7 and 8 at the 490 K. If yes, please mention it.

23) p18l8: "... one previous winter.". Please recall which winter it is.

24) P20l1-2: "This is particularly interesting given reported differences between years with early and late Arctic final warmings, which have not, in general, accounted for the suddenness of those final warmings (e.g. Waugh and Rong, 2002; Akiyoshi and Zhou, 2007);...". In recent studies on Frozen-In Anticyclones (FrIACs), tracer transport was linked to the suddenness/abruptness of final warmings (see e.g. Allen et al. (2012), Thiéblemont et al. (2013) or Thiéblemont et al. (2016)).

References:

Allen, D. R., A. R. Douglass, G. E. Nedoluha, and L. Coy (2012), Tracer transport during the Arctic stratospheric final warming based on a 33-year (1979-2011) tracer equivalent

latitude simulation, Geophys. Res. Lett.,39, L12801, doi:10.1029/2012GL051930.

Thiéblemont R., Y.J. Orsolini, A. Hauchecorne, M.-A. Drouin and N. Huret, A Climatology of Frozen-In Anticyclones in the Spring Arctic Stratosphere over the Period 1960-2011, J. Geophys. Res., 118, 1299-1311, doi:10.1002/jgrd.50156, 2013.

Thiéblemont R., K. Matthes, Y.J. Orsolini, A. Hauchecorne and N. Huret: Poleward Transport Variability in the Northern Hemisphere during the Final Stratospheric Warming simulated by CESM(WACCM), Journal of Geophysical Research, 121, doi:10.1002/2016JD025358, 2016.
* * *

---

## Short Comment (SC2) · 27 Sep 2016

This document is intended for readers who are non-specialist in dynamical systems theory and who are using results on Lagragian Descriptors. The document attempts to clarify some key aspects of this methodology which have been misinterpreted in recent literature. I adopt the format of Frequently Asked Questions, in order to trace an easy to follow path for these readers.

**1. What are Lagrangian Descriptors?**
These are functions obtained from particle trajectories which evolve advected by fluid flows according to a dynamical system. These functions evaluate from time, $t - \tau$, to time, $t + \tau$, the integral along the particle trajectory of a *positive* quantity such as: modulus of the velocity, modulus of the acceleration, modulus of the velocity or acceleration raised to specific powers, etc. One of these functions very frequently used, called function $M$, is the one used by Manney and Lawrence in the work under discussion, that considers the integral of the modulus of the velocity along the trajectory. It provides the arclength of the path traced by the trajectory.

These functions are useful because they highlight, by means of singular features, invariant stable and unstable manifolds of hyperbolic trajectories of the underlying dynamical system. These mathematical objects are of geophysical interest because they are related to transport barriers of purely advected fluid particles.

**2. What are singular features of the function $M$?**
Singular features of the function $M$ are described in Fig. 2 of the article by Mendoza and Mancho (Nonlin. Processes in Geophys. 2012) and in Figs. 10 and 11[1] of Mancho et al. (Comm. Non. Sci. Num. Sim. 2013) as abrupt changes in $M$ which are quantified by discontinuities in the derivative of $M$ along a specific direction crossing the manifold. The singular features in $M$ are aligned with the invariant manifolds.

For discrete dynamical systems Lopesino et al. (Comm. Non. Sci. Nume. Sim. 2015) have defined a different kind of Lagrangian descriptor, maintaining the idea of integrating positive quantities along trajectories, but which allows a rigorous treatment. There singularities are discussed in terms of undefined derivatives of $M$ at the points of the manifolds position, along lines which are transverse to the manifold.

**3. What is novel in the method of Lagrangian Descriptors with respect to previous work based on time averages along trajectories?**

Lagrangian Descriptors are based on integrals along trajectories. These can be converted into time averages by dividing by the time period of integration. Time averages have been related to phase space structures of dynamical systems through notions
* * *
[1]Fig. 11 has a typo in its caption. The caption should refer to Fig 10c) instead of 13c)

of ergodicity. A general framework that has been used in the context of fluid flows is the ergodic decomposition. This approach was developed in Malhotra et al. (Int. J. Bifurcation and Chaos 1998), Poje et al. (Phys Fluids, 1999), Mezic and Wiggins (Chaos 1999) and Susuki and Mezic (IEEE 2009) and is based on the fundamental work of Rokhlin (Am. Math. Soc. Transl. Ser. 1966). In particular Mezic and Wiggins (Chaos 1999) and Susuki and Mezic (IEEE 2009) highlight the importance of the Birkhoff ergodic theorem which states that in the limit $\tau \to \infty$ averages of functions along trajectories of measure preserving dynamical systems defined on compact sets do exist, and level sets of these limit functions are invariant sets. It should be noted that the Birkhoff ergodic theorem has *not* been proven for general velocity fields with aperiodic time dependence.

Results on Lagrangian Descriptors (LD) bring novel ideas with respect to these previous works:

• One novelty consists of the fact that LD are based on the integrals of *positive* quantities along trajectories forwards and backwards in time, while the work based on time averages considers the forward integration of *any* quantity along trajectories.

• A second novel aspect is that Mancho et al (2013) have shown that the integral of *certain positive quantities* (there exist positive quantities that fail in the goal) for sufficiently large integration time $\tau$, highlight invariant manifolds of hyperbolic points by means of singular features which are visible in the function $M$. The method of time averages does not visualize the phase space structures by means of singular features, but by means of level sets once the average has converged.

• These differences make it possible for LDs to visualize the invariant manifolds of a simple linear saddle, while the method of the time averages cannot. The reason for that is that the averages along trajectories of typical reported quantities, such as the horizontal component of the velocity, does not converge (in fact the trajectories do not remain in a compact domain, as required), and thus level sets have no dynamical

interpretation.

• A third novel aspect is that singular features of LD are visible in time aperiodic dynamical systems such as those found in geophysical flows and accurately represent invariant manifolds as confirmed by numerical simulations contrasted with other techniques. The visualization of these features is accurate even if the average along the trajectories has not converged. We note that the Birkhoff ergodic theorem has not been generalized to the case of aperiodically time-dependent vector fields, and thus in these cases where LD provide insights the method of time averages discussed in the literature does not.

There exist cases, however, in which LD brings no novel aspect with respect to time averages. For instance the analysis of the dynamical behaviour around a linear elliptic point by means of LD does not highlight any singular feature aligned with invariant manifolds of hyperbolic points as there are no hyperbolic points in this case. Time averages of positive quantities along trajectories converge in this case because trajectories remain in a compact domain. In this particular example, once the average has converged, level sets are in 1-1 correspondence with the trajectories. Since LD are related to the time averages by a constant factor, level sets of LD computed for the required integration period, are also invariant sets.

**4. What is the Objectivity property discussed in the literature? Is it important for practical purposes that LD satisfy that property?**

A scalar valued, time-dependent, function is said to be objective if it is invariant under Galilean coordinate transformations. In other words, the pointwise values of a function are the same at points that are transformed under a Galilean transformation, for each value of time. The function $M$ clearly does not have this property, and for this reason its utility for revealing phase space structure has been questioned in the literature (Haller, Ann. R. Fluid. Mech. 2015), although no examples of failure are pointed out.

The next example, however, shows that objectivity, understood as a property of functions which preserve pointwise values under a Galilean transformation, *is not a property that is desirable for any tool designed to reveal the phase space structure*, since the phase space structure may not be invariant under Galilean coordinate transformations.

For example, a system which is at rest, under a Galilean transformation having the form of a rotation with angular velocity $w = 1$, becomes described by the equations of a simple harmonic oscillator with both mass $m$ and constant $k$ equal to 1. The phase portrait of this system is described by concentric circles (1-tori) around an elliptic fixed point, and it is very different to the phase portrait of the system at rest consisting of a plane of fixed points. Therefore the $M$ function should provide different information in each case –information that reflects the phase space structure for the particular dynamical system. In the answer to question 3 we have already explained how $M$ recovers the phase portrait of a linear elliptic point. It is clear that if $M$ satisfied the criterion of objectivity it would be the same for both systems and thus it would not distinguish between the phase space structure for each of these very different systems.

**5. Do Ruiz-Herrera (Chaos 2015) results disqualify the use of Lagrangian Descriptors in Geophysical flows?**

Ruiz-Herrera (Chaos 2015, arxiv 2015) provides a different approach to the concept of singularity in $M$ to that referred in paragraph 2. He shows that in some specific examples in the limit $\tau \to \infty$, the function $M$ has no singularities -in the sense that he has defined- that highlight invariant manifolds. Balibrea et al (arxiv 2015) show, however, that singular features -in the sense introduced in the paragraph 2- are still present in Ruiz-Herrera examples.

Ruiz-Herrera' results (Chaos 2015, arxiv 2015) are not applicable beyond the hypotheses satisfied by his examples which, on the other hand, as detailed below, are applicable to a rather limited type of flows. None of the geophysical flows used in the Manney and Lawrence work, under discussion here, are similar in any remote sense to

those assumed by the theorems in (Chaos 2015). Thus inferring that those theorems prove something about the velocity fields considered in this work is an unsupported statement. Furthermore the debate about the capacities of $M$ for highlighting invariant manifolds in this open review manuscript is artificial simply because the authors use the function $M$ with a different purpose.

Ruiz-Herrera' results are for particle trajectories mainly in unbounded 2D flows, in which at least one of the velocity components of the trajectory is unbounded and grows much faster than the other. Some of these assumptions are of crucial importance in his construction. These type of trajectories and flows, however, are rather far from those found in atmospheric and oceanic flows, in which particle velocities are bounded and oscillating, remain in finite domains and velocity fields are bounded and oscillating. More specifically, Ruiz-Herrera' theorem III.1 from (Chaos 2015) uses a linear velocity in the y-component, and Theorem III.2 uses a linear velocity in the x-component and y-component in some piecewise defined domains.

The debate introduced by Ruiz-Herrera on the inability of the $M$ function to capture invariant manifolds and hyperbolic trajectories in the context of typical geophysical flows, solely based in his results, over exaggerates the scope of what he has actually done. Moreover, it contradicts plenty of published work in geophysical contexts showing numerical evidence which confirms the ability of $M$ to capture invariant manifolds highlighted as singular features by systematically contrasting the method with other techniques such as direct computation of manifolds, advection of blobs, Finite Time Lyapunov Exponents, Finite Size Lyapunov Exponents, and observations of drifters and balloon trajectories:

1. C. Mendoza, A. M. Mancho. The hidden geometry of ocean flows. Physical Review Letters 105 (2010), 3, 038501-1-038501-4.

2. C. Mendoza, A.M. Mancho. The Lagrangian description of aperiodic flows: a case study of the Kuroshio Current. Nonlinear Processes in Geophsyics 19, (4) 449-472

(2012)

3. de la Cámara, A., Mancho, A. M., Ide, K., Serrano, E., and Mechoso, C. R.: Routes of Transport across the Antarctic Polar Vortex in the Southern Spring, Journal of the Atmospheric Sciences, 69, 741-752, 2012.

4. E. L. Rempel, A. C.-L. Chian, A. Brandenburg, P. R. Munoz, S. C. Shadden, Coherent structures and the saturation of a nonlinear dynamo, J. Fluid Mech. 729 (2013) 309-329.

5. A.M. Mancho, S. Wiggins, J. Curbelo, C. Mendoza. Lagrangian Descriptors: A Method for Revealing Phase Space Structures of General Time Dependent Dynamical Systems. Commun. Nonlinear Sci. Numer. Simul. 18, (12) 3330-3357 (2013)

6. de la Cámara, A., Mechoso, C. R., Mancho, A. M., Serrano, E., and Ide, K.: Isentropic Transport within the Antarctic Polar-Night Vortex: Rossby Wave Breaking Evidence and Lagrangian Structures, Journal of the Atmospheric Sciences, 70, 2982-3001, 2013

7. C. Mendoza, A. M. Mancho, S. Wiggins. Lagrangian Descriptors and the Assessment of the Predictive Capacity of Oceanic Data Sets. Nonlinear Processes in Geophysics 21, 677-689 (2014)

8. V.J. Garcia-Garrido, A. M. Mancho, S. Wiggins C. Mendoza. A dynamical systems approach to the surface search for debris associated with the disappearance of flight MH370. Nonlin. Processes Geophys., 22, 701-712, 2015 (2015)

9. A. Guha, C. R. Mechoso, C. S. Konor, R. P. Heikes, Modeling Rossby Wave Breaking in the Southern Spring Stratosphere. J. Atmos. Sci. 73 (1) 393-406 (2016).

10. V. J. García-Garrido, A. Ramos, A. M Mancho, J. Coca, S. Wiggins. A dynamical systems perspective for a Real-Time Response to a Marine Oil Spill. Marine Pollution Bulletin (2016). DOI: 10.1016/j.marpolbul.2016.08.018

Finally it is worthwhile to emphasize, specially for readers in the atmospheric sciences, that the fact that *observed* balloon trajectories in the lower stratosphere (Refs 3,6), or in situ oil spill *observations* in the Canary Islands (Ref. 10) follow the geometric structures extracted with LDs, provide strong empirical evidence that LDs are useful to study transport in realistic atmospheric/oceanic flows.

Please also note the supplement to this comment:
http://www.atmos-chem-phys-discuss.net/acp-2016-633/acp-2016-633-SC2-supplement.pdf

---

## Short Comment (SC3) · 27 Sep 2016

**Response to: "Frequently asked questions about Lagrangian Descriptors" by Ana Mancho**

Alfonso Ruiz-Herrera*

September 27, 2016

Next I provide a reply to Mancho's comment in order to avoid misleading conclusions in the literature.Vague comments (without external peer review process) are not adequate to clarify a methodology.

**1   What are Lagrangian Descriptors?**

The $M$ function does not detect the invariant manifolds in many simple dynamical systems, e.g. $x' = x, y' = -2y$ or $x' = f(x), y' = -yf'(x)$ with $f(x) = \tanh x$. In fact, in a neighbourhood of the $y$-axis (the stable manifold of the origin) the contour lines of the $M$ function are horizontal lines in the previous systems.

**2   What are singular features of the $M$ function?**

Mancho, Wiggins, and their co-workers always use an unclear definition for the concept of singularity (even it looks rigorous). The method of Lagrangian Descriptors aims to detect mathematical objects with mathematical tools. The sentence: "Singular features are defined as abrupt changes in $M$ which are quantified by discontinuities in the derivative of $M$ along a specific direction crossing the manifold" is meaningless from a mathematical point of view. Please, give a formal definition. For instance, everyone knows that $f(x) = |x|$ is not smooth at 0 because $\lim_{x \longrightarrow 0} \frac{f(x)-f(0)}{x-0}$ does not exist.

Two remarks are in order:

- We have proved that the contour structure of the $M$ function has no dynamical significance in the detection of invariant manifolds (independently of the definition under consideration).

- As mentioned in my previous report, the theorems presented in ( *Lopesino et al. 2015*) are a consequence of the diagnostic itself because $MD_p$ is non-smooth if, for some iteration,

$$x_{i+1} = x_i \ \ or \ \ y_{i+1} = y_i. \tag{2.1}$$

*Departamento de Matemáticas, Universidad de Oviedo, Spain (alfonsoruiz@dma.uvigo.es, ruizalfonso@uniovi.es).

**3  What is novel in the method of Lagrangian Descriptors with respect to previous work based on time averages along trajectories?**

Analyzing the novelty of a diagnostic that leads to inaccurate responses does not seem very interesting. Mancho, Wiggins, and their co-workers have emphasized that their method is computationally cheap. However, in all global Lagrangian (diagnostic) computations of invariant manifolds in general data sets, the main computational cost from the advection of a large number of initial conditions. So the M function is no less (or more) computational than, say, the FTLE analysis for the same level of spatial resolution.

**4  What is the objectivity property discussed in the literature? Is it important for practical purposes that LDs satisfy that property?**

The notion of objectivity refers to the independence of the observer, see

- *Non-objectivity of the M function and other thoughts*, Interactive discussion in Nonlinear Processes in Geophysics about the paper *Detecting and tracking eddies in oceanic flow fields by Rahel Vortmeyer-Kley, U. Grave, and U. Feudel.*

Invariance under Galilean transformations is much weaker. However, I agree with Ana Mancho that the method of Lagrangian Descriptors is not objective (even in this weak sense). The message of Haller's comment was: The $M$-function is not objective. Mancho has now a different opinion on this because she stated in her reply to Haller's comment that the $M$-function was objective. ( See Remarks on the comment Non-objectivity of the $M$ function and other thoughts.) As emphasized in Haller's comment, one can simply point out that this diagnostic is not objective and hence cannot possibly to capture anything intrinsic about material transport. End of the discussion. We mention that Mancho's reply was posted without external peer-review process because of she is in the editorial board of Nonlinear Processes in Geophysics (the journal that contains most papers on Lagrangian Descriptors).

**5  Do Ruiz-Herrera (Chaos 2015) results disqualify the use of Lagrangian Descriptors in geophysical flows?**

As mentioned in my previous report, Ana Mancho and her co-workers in (Balibrea 2015) misrepresent what they have done. I have provided a detailed list of their contradictions. For instance, Figure 1 in *(Ruiz-Herrera arxiv)* is exactly the same as Figure 1 (c) and 2(a) in *(Mancho et al 2013)*. However, they have introduced the following misleading comment in *(Balibrea-Iniesta 2016)*, (see caption in figure 1):
*This figure should be compared with figure 1 of the comment of Ruiz-Herrera.*

The performance of the method of Lagrangian descriptors has been discussed theoretically just in the trivial system

$$\begin{cases} x' = x \\ y' = -y \end{cases} \tag{5.1}$$

In Mancho's applications and Manney-Lawrence work, the geophysical flows are not similar in any remote sense to (5.1). Of course, the performance in this trivial system is not enough to provide a effective diagnostic for any flow. It is clear that the $M$ function always creates patterns when plotted over the initial conditions. However, as emphasized in my work, this output has no dynamical significance.

The message of my work is that the method of Lagrangian Descriptors can fail in simple systems. There are many counter-examples to the method of Lagrangian Descriptors, for instance, $x' = f(x), y' = -y$ with $f$ bounded; $x' = 2x, y' = -y$ or $x' = f(x), y' = -yf'(x)$ with $f(x) = \tanh x$. More pathologies and counter-examples are discussed in (Ruiz-Herrera Chaos 2015) and (Ruiz-Herrera in press). Therefore, we can not expect reliable responses in complex systems.

As mentioned in my previous comment, the $M$-function was used for the first time in the pioneering work

- A.J. Jimenez Madrid and A.M. Mancho, *Distinguished trajectories in time dependent vector fields,* **Chaos 19** (2009), 013111.

In two papers published in the same journal,

- A. Ruiz-Herrera, *Some examples related to the method of Lagrangian Descriptors,* **Chaos 25** (2015), 063112,

- A. Ruiz-Herrera, *Performance of Lagrangian Descriptors and Their Variants in Incompressible Flows*, **Chaos** (in press),

I have shown that the contours of the $M$-function have no significance in the detection of barriers to transport (under any consideration). *Chaos: An Interdisciplinary Journal of Nonlinear Science* offers the possibility of comments to regular papers, see

- http://scitation.aip.org/content/aip/journal/chaos.

Mancho, Wiggins and their co-workers submitted a reply to *(Ruiz-Herrera 2015)* the last year. Please, submit your new critiques to the journal in order to avoid misleading conclusions in the literature.

---

## Author Response (AR1)

**General Changes**

In addition to the changes made directly in response to the referees' comments, which are detailed separately, we have made several other changes to improve the clarity/completeness of the presentation, primarily in the figures (accompanied by minor modifications in the text). We describe below the changes made to the figures, and the motivation for them. None of these changes alters our conclusions in any way; we hope that they improve the clarity of the presentation.

**Figure 1 (490 K Polar Processing Diagnostics)**
We have removed the timeseries of $V_{NAT}/V_{vort}$ (panel b in old figure) since the information provided in panel a (minimum temperatures) and Figure 3 lead to the same conclusions (this was suggested by referee #3). We have also simplified the sunlit vortex area panel (panel d in old figure; now panel c) by including the total vortex area only for 2015/16 (thin red line), and adding the daily climatological maxima of MERRA-2 vortex area (the topmost thin black line) to help make our point that the 2015/16 vortex was unusually large for most of the season.

**Figure 3 (Winter Polar Processing Statistics)**
We have removed the vertically summed number of days below PSC thresholds (panels b and d in old figure) since the information provided in those panels was mostly redundant with the combination of that given by the minimum temperatures shown in Figure 1a and the winter mean $V_{PSC}/V_{vort}$. We have also added error bars that represent the sensitivity of the winter mean volume diagnostics to the PSC thresholds used. The upper extent of the error bars represent winter mean $V_{PSC}/V_{vort}$ calculated using temperature thresholds with $T < (T_{PSC} + 0.5$ K), whereas the lower extent of the error bars represent the same but for temperature thresholds with $T < (T_{PSC} - 0.5$ K).

**Figures 9 - 11 (MLS and function M maps + M vs sPV scatterplots)**
We have done our best to increase the visibility of the continent outlines and latitude/longitude divisions on all the maps. To aid this on the function M maps, we changed the vortex edge PV contours on these maps to cyan (so as not to interfere with the gray continent outlines and white latitude/longitude sectors). We also change the PV contours overlaid on the function M maps (third row in each figure). In the old versions of the figures, we used contours from the 12UT MERRA-2 PV fields, which were slightly misaligned with the function M maps that use data from trajectories initialized at 00UT. As a result, we now show 00UT PV contours on the function M maps.

**Figure 14 (Multi-vortex time series)**

We have changed the colors to improve clarity, and to be consistent with the color changes to Figures 12 and 13 described in our responses to the reviewers. Now, the black, blue, and green lines correspond to the correct vortex regions in Figures 12 - 13, and the transients are properly labeled at both 490 and 550 K (before, the short-lived blue vortex at 550 K in late March was erroneously labeled "third"). The bulk quantity has also been changed from gray to orange to prevent confusion with the gray envelopes used for the black (parent) vortex.

We also changed the names of the vortices to use a consistent terminology throughout. We refer to the small offspring regions as offspring-p (green) for "persistent offspring" or offspring-s (blue) for "short-lived offspring". Offspring that persisted for less than about a day are referred to as "transient". The Figure 14 legend has been changed to reflect this.

Finally, the vortex-edge averaged windspeed panels (second row) now use quantities that are derived from an improved edge-following algorithm in CAVE-ART, which gives us a more accurate estimate of the average windspeeds and their variability around the edge (we have thus now included standard deviation envelopes on these lines). The differences between the old version and the new version are minor, and none of our conclusions or statements are affected by the change.

**Supplementary Animation 1 (CAVE-ART Identification of Vortices)**
We have increased the number of days that the animation covers to include all of 1 Feb through 30 Apr. We also changed the PV colormap and map projection (now orthographic) to better contrast regions with high and low PV. Furthermore, the vortex edge contours are now plotted using the CAVE-ART masks to omit the small high PV regions that exceed our sPV edge thresholds.

Equivalent ellipses are now colored to be consistent with Figures 12 - 14 in the main paper. Finally, we inverted the colors of the background and text so that the background is black and the text is white (to be more viewer friendly).
*This paper is a well-written, comprehensive study of the fate of the northern hemispheric polar vortex in spring 2016. The paper is well-structured into an Introduction, a Data and Method section, overviews the 2015/2016 polar vortex evolution in Section 3, and, finally, focusses on the early vortex breakup in March 2016 and the subsequent mixing of vortex air with mid-latitude air. Section 5 summarizes and concludes the paper with clear statements and with a friendly wink and hint to follow the NH vortex evolutions in the future with care. I'm sure, the authors will do so as they possess the suitable diagnostic tools to analyze global satellite and meteorological data in an efficient way.*

*The Arctic winter 2015/16 was extraordinarily cold and the vortex-wide temperatures felt as low that the conditions in January resembled those of the Antarctic in terms of chemical composition and PSC appearances. Besides the fascinating subject of the recent winter, it is the clarity in structure and writing which make the paper a joy to read.*

*The Introduction sets the scene by stating "that SSWs affect Arctic lower stratospheric chemical ozone loss in ways much more complex than a simple association of low (high) temperatures with more (less) ozone loss". So it is consequent to read the motivation as: "Thus, understanding the complex relationships between SSW dynamics, stratospheric vortex evolution, and chemical composition and processing, is critical to diagnosing and predicting ozone loss and recovery in the Arctic and its climate consequences." After a short historical view, the authors come back to the topic by discussing the interannual variability of NH winters (minor/major SSWs and dates of final warmings) in close relation to the chemical ozone loss. Some of the main results are already anticipated in the fourth paragraph: (1) "the 2015/16 Arctic winter was the coldest on record (since at least 1979)"; by the way, Matthias et al. (The extraordinarily strong and cold polar vortex in the early northern winter 2015/16, GRL, under review) showed that it is was indeed the coldest in the recent 68 years. (2) a major final warming "beginning in early March 2016 resulted in the breakup of and dispersal of chemically processed air from the vortex, which halted chemical loss much earlier than in 2011".*

*Section 2 reviews the data sources and the methods. It is impressive to see that the*

*latest versions of MERRA-2 and MLS data are used. The diagnostic quantities and tools are presented systematically. Even newer developments which are in the pro-cess of publishing are explained in a comprehensive way. The authors apply a broad spectrum of well-established and newly developed diagnostics to quantify the spatio-temporal variation of the various trace gases and dynamical quantities specifying the mixing in the surf zone of the polar vortex.*

*The Overview of the 2015/16 vortex evolution and composition focusses on thermal and chemical aspects. For a reader not so familiar with all the peculiarities of the previous winters less direct comparison would be advantageous. Maybe, sentences like (around line 320) "Ozone continued to decrease in the vortex at a rate slightly faster than that in 2011 until the beginning of March 2016. If uninterrupted, ozone values would have been expected to drop lower than those in 2011 by mid-March." could be slightly reformulated to give for example explicit values of rates. But this, for sure, is only a matter of taste. At the end, main results for the winter 2015/16 (in relation to previous ones) are presented and key words are: "leading to unanticipated extremes in Arctic polar processing, the 2015/16 winter stands out as yet another unexpected extreme in variability of the Arctic winter stratosphere." , "The period of over a month, from late December through early February, with temperatures below the ice PSC threshold was unprecedented in the Arctic", "much greater degree of dehydration", and "extreme denitrification", "extensive early winter chlorine activation", and "chemical ozone loss began early". And finally, we read: "Thus, the critical factor resulting in less ozone loss than in 2011 was the much earlier increase in temperatures and vortex breakup in 2016."*

*The Section 4 about the 2015/16 major final warming and the resulting vortex breakup and mixing exemplifies the trace gas evolution and the strength of the transport barrier, and the mixing especially in the surf zone by refined analyses at different isentropic surfaces representing the conditions in the middle and lower stratosphere. I'm not the expert to evaluate the details of the applied diagnostics but the text reads logical and the conclusions are based on well-funded results from the respective simulations. Altogether, the paper can be published in the present form!*

**Initial Reply, Posted 7 Sep 2016**

We thank Dr. Andreas Dörnbrack for his very positive review, and are glad that he enjoyed reading the paper -- we certainly enjoyed writing it, and are pleased to hear from someone who

appears to share our fascination with the unpredictable wide range of variations in Arctic stratospheric meteorology.

With regard to Section 3, the overview of the 2015/2016 vortex evolution and composition, we do appreciate that this material could be a little hard to follow for the reader who is not already familiar with the conditions and amounts of ozone loss in other recent winters.  When we revise the paper, we will work on being more explicit about the conditions and ozone loss amounts / timing in the previous winters that are highlighted, so it is easier for the reader to follow the comparisons.

**Added author comment, 26 Oct 2016:**

In addressing the other referees' comments, we have made several clarifications to Section 3, primarily to note more explicitly the connections between the meteorological diagnostics shown in Figures 1 and 3 and the trace gas evolution shown in Figure 2, and to make the motivation of the comparisons with the previous winters clearer.  We believe this last point will help to address the slight confusion Dr. Dörnbrack indicated in the comparisons with other winters.
*This paper is certainly comprehensive, appropriate for ACP and most probably, correct. It was, however, also difficult to read and review. It comes across, at least to this reviewer, as a so-called "core dump" of information. As a consequence, I readily admit that this review is probably incomplete and that I probably missed pieces of information that to the authors, at least, they would deem critical. My comments are therefore (with one exception) more editorial than scientific.*

We thank the referee for their valuable comments regarding our manuscript. As we describe below and in our responses to the other reviewers, we have made numerous modifications that we hope clarify our paper and elucidate how the information we have included is focused on a comprehensive analysis of the major final warming in 2016.

*General 1. One general science question that I think could use a greater exposition is the question of the MFW. The authors imply that this hybrid event in 2016 is unusual. Some context as to its occurrence frequency would be helpful. Is this the first to occur in the AURA record?*

This event was not the first in the Aura record; the 2004/2005 Arctic winter was similar in that its unusually cold conditions were cut short by an early final warming around 10-12 March 2005. However, 2015/2016 was much colder, and the potential for polar processing more severe. In addition, final warming events as early as these are uncommon. According to the table provided in Hu et al., 2014, only 13 winters between 1958 and 2012 had early final warmings before 1 April, of which only 5 occurred before March 15. We have included additional text in our conclusions section that discusses these points, including differences and similarities in the 2005 and 2016 events.

*2. As far as presentation, Section 3 is a case in point as exemplifying my concerns. Figures 1-3 are introduced in rather random order, with lots of information, which, while not technically wrong, may well be irrelevant. The authors present a whole bunch of figures (14 panels in all for Figures 1-3) and then jump back and forth in a scatter shot discussion. This is very taxing to read. The first three paragraphs do not even discuss the 2015-16 season, but rather present a literature review of 3 three previous winters.*

*Line 27 on page 7 is a good example. The statement is simple- temperatures in a particular year (a year which was not the subject of the present paper) were cold enough to activate chlorine for a prolonged period of time. So why do we need to refer to four separate figure panels (Figure 1a and b Figure 2d and e?) to make this simple point (which again is irrelevant to the subject of this paper that is nominally about 2016)? In fact, I don't understand why Figure 2 is referred to here. Is it because ClO was going up? That is not explained.*

We have made several revisions to section 3, including adding text to clarify that we are discussing Figures 1 and 2 together so that we can draw the direct connection between the meteorological diagnostics and the implications of their evolution for changes in the trace gases (e.g., minimum temperatures are directly related to the evolution of HNO3 and H2O via PSC formation and denitrification/dehydration; these are in turn linked to chlorine activation and deactivation; sunlight exposure shown in Figure 1 is directly linked to elevated ClO and ozone loss shown in Figure 2; etc).  By discussing these figures in a unified way, we elucidate the dependences of the composition on the meteorological conditions.  We have also simplified Figure 1 by removing the $V_{NAT}/V_{Vort}$ panel, which the referee points out below provided little additional information, and reducing the number of lines on the sunlit vortex area panel (now Figure 1c).

*3. Adding up all the panels in 15 figures, the paper contains 128 separate graphs.  I confess that I found it difficult to subject each and every one to the scrutiny they probably deserve; I do nonetheless strongly suspect that they are not all necessary.  As an example, I did examine one specific panel- that of Figure 1b. All references to Figure 1b occur with a simultaneous reference to Figure 1a. I therefore conclude that Figure 1b can't be necessary since it never is referred to independently of Figure 1a.  So it should be deleted. Especially since they never describe it (what is V_nat/V_vort?-they briefly mention it on page 9, but not in the context of Figure 1).*

We have reduced figures and figure panels where possible, including removing the panel in Figure 1 that the reviewer mentioned as a candidate for deletion.  In addition, we have removed two panels of Figure 3, all of Figure 12 (four panels), and four panels in each of Figures 13 and 14 (now Figures 12 and 13). We do stress, however, that one of the main points of our paper is that there is good agreement between the dynamics (represented by diagnostics derived from MERRA-2) and chemistry/transport as seen by measurements from MLS, which all paint a consistent picture of what happened throughout the 2015/16 Arctic winter season, and in order to make this point, we need to show these fields and their evolution.  Every figure panel that is included in the paper is used to support some point; a majority of them are referred to not only when initially

discussed, but also referred back to to support points made about or show consistency with succeeding figures.

*4. I also think Figure 3 is unnecessary. Not that it's technically incorrect, but it adds no new information that is not conveyed in Figures 1 and 2. Indeed, their concluding sentence on lines 19-20 of page 9 can easily be gleamed from Figures 1-2.*

As per our reduction of figure panels mentioned above, we have removed the panels showing the vertically summed number of days below the PSC thresholds. We now only show the winter mean $V_{NAT}/V_{Vort}$ and $V_{ice}/V_{Vort}$.  Although there is some overlap with the information conveyed in Figures 1 and 2, Figure 3 is still necessary to show that the winter mean polar processing and ozone loss potential in 2015/16 was unusually large, comparable to that in 2011; this point cannot be seen by looking at Figure 1 alone.

*Minor*

*1. Figures 5-14 (with the exception of Figure 8) are essentially 3 sets of three figures for 850, 490, and 550 potential temperature surfaces. It would be introductory few sentences at the beginning of Section 4 explaining why they chose these three levels. Even if it was empirically determined that they were good representative levels, they should at least say that. As it reads, it just says (for example) Figures 5-7 without telling the reader where you are going with this. You have to read almost 2 pages of the draft before you find out that these 3 figures are for three separate altitudes.*

At the end of section 3, we have a transition paragraph that points out the three levels we are going to focus on in section 4 and their significance.  In addition, we have added text at the beginning of section 4 noting that parallel figures are going to be shown at three levels to contrast/compare their behavior.

*2. Figure 6 vs Figure 7. If I understand correctly, the text on page 11 suggests (line 25, compared with line 7) that one difference is that N2O and O3 do not show mixing out of the vortex at 550K but they do at 490 K. Looking at Figures 6c and 7c, I see no difference. Am I supposed to?*

We apologize for a lack of clarity in these statements.  The main point in the paragraph on 550K was the consistency of persistent strong trace gas gradients along the vortex edge with the stronger persistent transport barrier seen in PV gradients and $K_{eff}$ at this level.  This does indeed imply less mixing out of the vortex at 550K than at 490K.  We have revised the text to clarify both points.

*3. Figure 1c: what is Max PVG? Those three letters do not appear anywhere else in the text or figure captions.*

We apologize for this oversight; max PVG stands for "maximum PV gradients." We have changed the text in the Figure 1 caption to read: "maximum gradients of scaled potential vorticity as a function of EqL (Max PVG) ..." to make this clear.

*4. Figure 15: What do the colors mean? There is a label that says "first", "second" etc, but doesn't explain what those terms mean other than "bulk". Are they related to the colors of various fragments in Figure 14. If so, it should say so.*

We have clarified our references to the colors and the regions in the text. Rather than referring to the offspring regions as first, second, etc, we now refer to them as "parent", "offspring-p", and "offspring-s" to describe the amount of time these regions existed (the "p" and "s" in offspring-p and offspring-s stand for "persistent" and "short-lived", respectively). Vortices that persisted for about a day or less are labeled "transient". We have also added text to the Figure 14 caption to explain these names/references, and they are explained in the text where they are first introduced (in conjunction with Figures 10 and 11).

*5. Abstract: Line 20. Where do they show chlorine in the offspring vortices? Figure 15 does not show chlorine. There are cryptic references to chlorine activation and deactivation scattered throughout the text, but I could not find where it pointed to a figure saying "this shows the deactivation of chlorine etc. etc."*

We note explicitly in the text related to Figure 14 regarding shorter lived species that the average values of ClO, HCl, and HNO3 are very similar across the parent and offspring vortices, and hence showing the evolution in individual vortices (as in Figure 14) would only add panels without adding information. We have added text explicitly noting that how ClO evolves in each vortex can be deduced from Figure 2e. Also, the time evolution of ClO is shown clearly in Figures 4e and 6e.
*This paper provides a thorough description of the evolution of the 2015/2016 Northern Hemisphere stratospheric winter until the breakup of the polar vortex and the dispersal of its fragments. This winter was unique: while it initially presented the characteristics for an unprecedented ozone loss (i.e. prolonged temperatures below ice polar stratospheric cloud thresholds), an anomalous early and strong major final warming interrupted the ozone depletion process. Dynamical and chemical processes are characterized using Microwave Limb Sounder satellite trace gas measurements, and advanced mixing and polar vortex diagnostics derived from meteorological reanalysis. This case study of the winter 2015/2016, and its comparison with the series of singular recent winters, very well illustrates the complexity of the dynamical and chemical interactions that drive Arctic ozone depletion. In my opinion, this paper is important as it further contributes to showing that each Arctic winter season is unique and that substantial research efforts are needed to better understand their extreme variability and the consequences of this variability (e.g. on ozone depletion, stratosphere/troposphere couplings). The methods and diagnostics used in this study are scientifically sound and relevant. The analysis is very carefully conducted. My main criticism rather concerns the form: the main text and its figures are extremely dense and contains a lot (too much?) of information so that it is sometime hard to differentiate what is important from what is more anecdotal. While in some places the degree of detail seems to me exaggerated (e.g. p10l32-p11l10 where tracer extrusions are discussed while not really obvious), in other places, including further details may help to make the paper easier to follow (see comments below).*

Motivated both by this comment, and those of Referee #3, we have gone through the paper with a focus on assessing the clarity and necessity of the text and each figure/figure panel. As a result, we have eliminated several figure panels that were not as critical to our message, and clarified the text to indicate the motivation for showing the information that is included.

Regarding the particular example given above of the discussion of filamentation in relation to Figure 6, we have revised and reduced this discussion to eliminate details that are less critical to the paper.

*Hence in my opinion, this paper is suitable for publication in Atmospheric Chemistry and Physics after consideration of the specific (minor) comments and suggestions provided below.*

We thank Dr. Thiéblemont for his careful and thorough review, and very helpful comments on our paper.

*Specific comments:*
*1) p3l32: Typo change "MERRRA" to "MERRA"*

Done.

*2) p4l20: Please provide further detail on the way the potential vorticity is scaled. The sPV is widely used throughout the paper so few precisions about it may be useful for the readers.*

Changed to "... (sPV, scaled to have a similar range of values throughout the stratosphere using a standard atmosphere value of static stability, as in Dunkerton and Delisi, 1986; Manney, et al, 1994) ..."

*3) Diagnostics (i.e. sections 2.3 & 2.4): This paper makes use of a very high number of diagnostics to describe mixing processes, transport, vortex size and so on. Although the different diagnostics are very well explained in the main text, non-expert reader may quickly be lost once the description of the (dense) analysis begins. The authors may consider adding a table which gives a summary describing (briefly) the different diagnostics and their usefulness.*

We have added a list that summarizes the transport and mixing diagnostics we use at the end of section 2.4. While we did not do the same for the polar processing diagnostics in section 2.3, we did remove some of the diagnostics previously shown in Figures 1 and 3, and added more explicit text clarifying how each diagnostic is related to the evolution of trace gases in the polar vortex.  We hope this helps reduce the complexity, and makes the motivation for including each diagnostic clear.

*4) p7l25: "The 2010/2011 winter".  Please mention the associated color line in bracket to help the reader.*

A note with the line colors has been added the first time each year is mentioned.

*5) p8l1: "In early January 2013". Same here, please mention the associated color.*

A note with the line colors has been added the first time each year is mentioned.

*6) p8l2: "strongest "vortex-split" SSWs on record" What does strong mean here? What defines the strength of a SSW (persistence, temperature, vertical extension?)? Please clarify.*

We have revised the text to indicate that we mean among the largest abrupt temperature increases, deepest range of wind reversals, and most prolonged periods of easterlies.

*7) p8l6: "2014/2015". Please mention the associated color.*

A note with the line colors has been added the first time each year is mentioned.

*8) p8l6: "brief minor SSW". Please give the date. (I guess early January)*

The date has been added: "...very brief minor SSW (with a brief vortex split on 5~January~2015)..."

*9) p8l13: Typo change "though" to "through"*

Done.

*10) p8l14: "unprecedented". On MERRA record? Please clarify.*

We now specify that it is in the MERRA-2 record: "...unprecedented in the Arctic, where the MERRA-2 record rarely shows more…"

*11) p8l31: "in 2015/1016, 2012/2013, and 2010/2011". Why not 2014/2015? The green curve looks similar in early winter on Figure 2a.*

This is indeed true if the period is limited to early winter, and we now simply state that it was similar in all the years highlighted.

*12) p11l1-2: "This is consistent [...] anticyclone during this period". Does anticyclone*

*refer to the Aleutian High here? Please clarify.*

We have modified this to note that we do, indeed, mean the Aleutian anticyclone.

*13) Figure 8: Please replace y-axis "Effective diffusivity" by "Keff" to be consistent with the main text (p11l30).*

We have labeled the y-axis in Figure 8b, as well as the color bars in Figures 5--7b, "Effective Diffusivity ($K_{eff}$)".

*14) p11l33: "Keff and M minima" Is it not rather M maximum? M maximum -> vortex edge -> transport barrier.*

Thanks for catching this error. We have revised it to say "...sPV gradient and *M* maxima, $K_{eff}$ minima, and strongest trace gas gradients…"

*15) Figure 9-14: Please make the continents more visible on maps and provide at least on longitude coordinate. Otherwise it is quite hard to follow Figures together with the main text and the geographical location that are refereed (e.g. Alaska p13l7 but also at other places).*

We have done our best to make the continents and latitude/longitude lines more visible in all cases. In addition, we have provided an orientation reference in the first figure caption for each type of map, noting that 0 degrees longitude is at the bottom of the maps and 90 degrees E to the right.

*16) P13l12: "in the anticyclone.". Is it not "in the edge of the anticyclone" that the M values are the strongest?*

It is both along the edge and along the persistent filaments that spiral into its interior; the text has been modified to reflect this.

*17) At 550 K, a doubled vortex edge appears in the main vortex fragment (see Fig 11, 14) from beginning of April. Is this an artifact or a real structure? Please comment on this.*

It is difficult to say whether the doubled-edge structure is real or an artifact since it represents the sPV dropping slightly below our vortex-edge threshold in the core of the fragment. However, we do not think it has any particular significance in relation to

transport/mixing in this case, since the M maps show that the highest M values are around the outer edge. We have added additional text stating this.

*18) Figure 13-15 (and associated text). The green and blue offsprings seem actually switched between the 490 and 550 K levels. If indeed this is the case, it may be confusing. Therefore, it may be more relevant to keep the same color for the upward extension of the same offspring.*

We have switched the colors of the vortices so those of the smaller offspring that persist longest (and are the extension of the same vortex in the vertical) are the same color. We now label the vortices as "parent," and "offspring-p" and "offspring-s" for the "persistent" and "short-lived" small vortex regions, respectively. (Offspring vortices that persisted about a day or less are labeled and described as "transient".)

*19) p15l17-19: May this vortices coherence dependence with height be partly related to differences in diabatic processes with height?*

We do believe this to be the case. There is a large body of literature showing that most final warmings proceed from the top down, and this is largely related to shorter radiative timescales in the middle to upper stratosphere. However, in removing material that, though interesting, seemed peripheral to our primary focus, we have deleted the statement that raised this question; therefore we have not modified the text in this regard.

*20) p16l7: "begins dropping earlier,": earlier than when? Please clarify.*

We meant "begins dropping earlier than the vortex area…" and have added this to the text.

*21) p16l7: "period between the beginning of the MFW and the split": is it the period between the two dashed lines? Please clarify.*

Yes, we have added a note saying this in the text: "...period between the MFW and the split (between the two vertical dashed lines in Figure 14)..."

*22) p16l19-21: "In fact, as seen in Figure 13, a coherent mass of air from the blue vortex persisted into April – represented in Figure 15 by the individual purple points labeled "transient". I guess these transient vortices are those seen in the supplementary animation and labelled 4, 5, 6, 7 and 8 at the 490 K. If yes, please mention it.*

You are correct, and we have added text pointing this out and referring to the animation in the discussion of Figure 14 (was Figure 15).

*23) p18l8: "one previous winter.". Please recall which winter it is.*

We have added this information: "...one previous winter (2012/2013, Figure 2d--f)..."

*24) P20l1-2: "This is particularly interesting given reported differences between years with early and late Arctic final warmings, which have not, in general, accounted for the suddenness of those final warmings (e.g. Waugh and Rong, 2002; Akiyoshi and Zhou, 2007);". In recent studies on Frozen-In Anticyclones (FrIACs), tracer transport was linked to the suddenness/abruptness of final warmings (see e.g. Allen et al. (2012), Thiéblemont et al. (2013) or Thiéblemont et al. (2016)).*

This is a very good suggestion; indeed we were remiss in not mentioning these studies. We have added a brief discussion of FrIACs following sudden/abrupt final warmings in this paragraph.

We would like to thank Drs. A. Ruiz-Herrera and A. M. Mancho for their comments regarding Lagrangian descriptors and the Function M (hereinafter referred to as M). Clearly there is an ongoing discussion about cases for which Lagrangian descriptors (including M) are applicable diagnostics for elucidating flow characteristics. Although neither Dr. Ruiz-Herrera nor Dr. Mancho have suggested changes to our manuscript, we would like to expound upon our use of M, and specifically why we think it provides useful information beyond that given by the other transport barrier/mixing diagnostics used in our analysis of the 2015/2016 Arctic winter.

We have applied M to the stratospheric polar vortex, which is a well-understood system in the atmospheric sciences. There is a large body of literature that has established that the polar vortex edge, or polar night jet, acts as a significant transport barrier that dynamically and chemically separates intra-vortex air from extra-vortex air (e.g., Schoeberl et al., 1992). Maps of M in the polar winter stratosphere highlight this dynamical separation. Because high/low values of M represent the "long/short" distances traveled by parcels advected by the flow, we know that the band of high M values in the vortex edge region represents the position, strength, and approximate width of the polar night jet (see, e.g., Fig 9 in our paper). This band separates two regions with low M values -- the intra- and extra-vortex air. As the polar vortex weakens and shrinks, M values in this band are reduced to the point where there is no qualitative separation that significantly distinguishes intra- and extra-vortex air (e.g., in Fig 9, 4th column, 3rd row).

Furthermore, when binned as a function of potential vorticity (PV)-based equivalent latitude (EqL), M agrees well with other instantaneous transport barrier/mixing diagnostics. Maxima in M as a function of EqL correspond well with maxima/minima features in PV gradients, trace gas gradients, and effective diffusivity (see Fig 8 in our paper). This is no surprise since, as explained above, the largest M values occur in the vortex edge region where PV gradients are largest. Hence, M is at least as useful as these other instantaneous diagnostics, even though it incorporates 30 days' worth of flow information (when using tau = 15 days).

Having established this agreement, M in addition allows us to examine local changes in transport and mixing that the other diagnostics, which are inherently calculated as averages around EqL contours, do not. And this in particular is what we find to be most useful about M in our context; it incorporates a history of the underlying dynamics. Investigation of instantaneous wind fields alone, for example, results in qualitatively similar maps to those of M, but M highlights the dynamical significance of short and long-lived features such as the vortex edge and vortex filaments. For example, parcels initialized within the vortex edge region obtain large values of M because, in the reverse trajectories, they very likely *originated in* the vortex edge, and in the forward trajectories, very likely *stay in* the vortex edge. Similarly, parcels initialized or drawn into a vortex filament obtain relatively large values of M because, in the reverse trajectories, they also very likely originated in the vortex edge -- but they don't reach vortex-edge levels of M because at some point they are drawn off and stirred out in the surf zone.

Our simple/visual "distance travelled" interpretation is largely qualitative, but we think it is intuitive and helps prove our point that the 2015/2016 Arctic polar vortex decay was unusually intense and rapid, especially in light of the unusual strength and size of the vortex before the major final warming. As Dr. Mancho pointed out, we are using M in a different manner, in a way that we think does not depend on the mathematical rigour necessary for discussing flow manifolds, hyperbolic trajectories, etc.

We have included in our manuscript additional references to the literature highlighting the use and criticisms of Lagrangian descriptors pointed out by Drs. Mancho and Ruiz-Herrera. Furthermore, we have added text to our manuscript to clarify our use of M and why the ongoing discussion surrounding Lagrangian descriptors does not affect our simple use of M.

**References**

Schoeberl, M. R., L. R. Lait, P. A. Newman, and J. E. Rosenfield (1992), The structure of the polar vortex, J. Geophys. Res., 97(D8), 7859–7882, doi:10.1029/91JD02168.

[revised manuscript text omitted]

---

## Referee Report (RR1)

Review of "The major stratospheric final warming in 2016: Dispersal of vortex air and termination of Arctic chemical ozone loss" by Manney and Lawrence

The revision has adequately addressed all of my comments and I found the clarity of the paper well improved. I thus recommend publication. Note just one typo at my last name in the acknowledgments (Thiéblemont instead of Thiémblemont).